# Episodic Knowledge Binding: A New Challenge for LLM Continual Learning

## Abstract

Large language models (LLMs) excel at learning individual facts but fail at a fundamental aspect of human cognition: binding related episodes through shared elements. Unlike humans, that effortlessly retrieve all encounters with a person or visits to a location after learning each separately, we demonstrate through controlled experiments that LLMs trained on single-event question-answering pairs cannot generalize to exhaustive multi-event retrieval. We formalize Episodic Knowledge Binding as the challenge of retrieving multiple related episodes when training lacks explicit multi-event supervision.

Differently from catastrophic forgetting, where models lose previously learned information, this binding failure persists even when training on aggregated data without temporal confounds, showing that models do not spontaneously develop multi-event retrieval from separate training points. Leveraging synthetic episodic narratives, we reveal a consistent binding gap across model scales (3B–13B and GPT-4.1) and narrative lengths (10–100 events): models attain high accuracy when entities appear in single events, but performance collapses when multiple related episodes must be retrieved. We find that (unsurprisingly) binding becomes harder with more events and that model scaling (more surprisingly) offers only minimal relief within our tested range. To address this problem, we propose Generative Cued Replay (GCR), that (i) inherently operates in a continual learning manner and, inspired by hippocampal memory consolidation, (ii) queries the model's parametric memory for related episodes when processing new events, (iii) synthesizing multi-event training data without storing past episodes at each new training step. This approach significantly improves binding without architectural changes, offering a practical method compared to exhaustive multi-event supervision which is both computationally infeasible as well as inherently more rigid. We release our Episodic Knowledge Binding benchmark to enable future research on this fundamental capability that LLMs are currently lacking.

## 1 Introduction

Human episodic memory does not just *store* events, it *binds* them through shared elements (Tulving, 1983; Tulving et al., 1972; Horner et al., 2015): when encountering a colleague at a conference, our hippocampus does not merely encode this new episode, it activates an entire network of related memories through common threads, the person, the topic, similar venues in a process known as pattern completion (Horner et al., 2015). This *binding* capacity, fundamental to human cognition, enables us to answer "When did I meet Sarah?" with a complete list of episodes, not just the most recent encounter.

Unlike semantic memory, which stores general facts, episodic memory is inherently relational, grounding experiences in time, space, and entity-specific details (Tulving et al., 1972). This remarkable integration is mediated by the hippocampus, which acts as a dynamic index for episodic memories Teyler & DiScenna (1986) and a pattern completion engine (Horner et al., 2015). At this index, specialized neurons, place cells that encode specific locations (Moser et al., 2008), time cells that segment temporal sequences (MacDonald et al., 2011), and concept cells that respond to entities regardless of modality (Quiroga et al., 2005), create a multi-dimensional binding space. The hippocampal indexing theory (Teyler & DiScenna, 1986; Teyler & Rudy, 2007) posits that the

hippocampus stores compressed representations that serve as pointers to reactivate distributed neocortical patterns during recall. Critically, this biological system does not just prevent forgetting; it actively connects related episodes, enabling mental time travel and coherent narrative construction (Schacter et al., 2007).

Anecdotal evidence shows that modern LLMs exhibit intriguing contradictions in this domain. The Reversal Curse demonstrates that models trained on "A is B" often cannot infer "B is A" (Berglund et al., 2023), suggesting rigid, unidirectional storage rather than flexible binding. Yet paradoxically, the same models can compose facts learned separately, inferring a city's identity from distances to Rome and London learned in different LoRA modules (Treutlein et al., 2024). These paradoxes suggest that parametric memory in LLMs, differently from hippocampal indexing, can cope with partial pattern completion while failing at exhaustive retrieval.

We study this binding challenge through controlled experiments with single-event QA training and multi-event QA testing, a setting that makes the problem clearly observable and measurable. While one could theoretically generate multi-event training data, this becomes impractical at scale: without ground truth episode structures, identifying and linking all related episodes across large corpora exceeds current capabilities. Even with frontier LLMs, context limitations and the combinatorial explosion of possible multi-event questions make exhaustive coverage infeasible. The binding problem manifests in both static training (where all data is available) and continual learning (where episodes arrive sequentially), though it becomes particularly salient in the latter where models are rarely exposed to related episodes simultaneously.

The field has focused on catastrophic forgetting, i.e. hoping that models remember that Paris is in France after learning about London (Gupta et al., 2024). In classic neural networks, techniques like elastic weight consolidation (Kirkpatrick et al., 2017), experience replay (Rebuffi et al., 2017b; Lopez-Paz & Ranzato, 2017b), and progressive neural networks (Rusu et al., 2016) aim to avoid such forgetting. However LLMs introduce a new and even more important challenge: i.e., how to effectively *bind* episodes through shared elements, as opposite as just *remembering* individual episodes – a challenge which we here term *Episodic Knowledge Binding*, and which remains largely unexplored (cfr related work in Sec.6 and Appendix: D) despite being fundamental to both human cognition and practical applications.

In this work, we study this challenge through a specific lens: after training on single episodes individually, can models retrieve all episodes matching a given retrieval cue purely from parametric memory? We design (Sec.2) controlled experiments with synthetic episodic narratives where each event is a single sentence encoding time, space, entity, and content. Models are trained on single-event question–answer pairs (e.g., "Where was Mary on March 5th?" → "Tokyo") but evaluated on multi-event retrieval (e.g., "List all times Mary was in Tokyo" → "Mar 5th; Sep 12th; . . . ").

Controlled setting allows to isolate and study the binding phenomenon: our systematic evaluation reveals a universal binding gap across scales (Sec. 3). Models of any tested scale (from 3B to 13B parameters, as well as GPT4.1), consistently fail at multi-event retrieval: they achieve high accuracy when an entity appears in exactly one event, while performance collapses as the number of matching events increases. This holds both when training sequentially (mimicking continual learning) as well as on aggregated data (removing temporal confounds and catastrophic forgetting effects): in other terms, fine-tuning on single-event QAs creates lookup tables, not episodic indices.

To address this gap, we propose (Sec.4) and evaluate (Sec.5) a novel approach inspired by hippocampal memory consolidation where new experiences trigger replay and rebinding of related memories, that we refer to as *Generative Cued Replay (GCR)*. When encountering a new episode about entity $ent$, GCR: (i) mimick pattern completion by querying the model to recall prior episodes involving $ent$ and (2) creates synthetic multi-event QAs combining recalled and new information. This biologically-inspired approach, requiring no architectural changes, significantly improves binding by progressively building parametric indices through rehearsal, similar to how cued recall spontaneously reminds of related episodes or later sleep-dependent consolidation strengthens episodic associations in biological systems Rasch et al. (2007).We summarize our contributions as follows.

**A new fundamental challenge revealed through controlled experiments**: We formalize the new challenge of Episodic Knowledge Binding, as retrieving multiple related episodes after learning them individually: this challenge is distinct from catastrophic forgetting, and we show it to be a severe limitation in current static as well as sequential training paradigms.

**Systematic evidence of binding failure**: Across model scales (3B-13B) and event lengths (10-100 events), we demonstrate that single-event training fails to induce multi-event retrieval capabilities, although a partial successful binding appears with small narratives of 10 events.

**Human-inspired approaches**: We propose Generative Cued Replay (GCR) strategies guided by biological memory consolidation principles, which we show to significantly improve binding without requiring direct multi-event supervision.

**A reproducible benchmark**: We release our synthetic episodic narrative generation code and evaluation framework to enable future research on this fundamental and new capability.

We believe episodic knowledge binding represents a new frontier in continual learning, beyond only preventing forgetting. For LLMs to succeed in tackling complex sequential tasks, the ability to bind related episodes through shared elements becomes essential. We believe our work establishes both the challenge and initial approaches, opening a new research direction at the intersection of continual learning, memory systems, and neural episodic representation.

## 2 PROBLEM FORMULATION

We adapt the episodic memory benchmark from Huet et al. (2025) to study a new challenge in parametric LLM training: *can models bind related episodes learned separately into queryable parametric indices?* The benchmark by Huet et al. (2025) provides a controlled framework for generating synthetic episodic narratives and evaluating memory recall through cue-based retrieval. We modify their approach in two key ways to focus on the binding problem. First, we focus on parametric memory, distinguishing between single-event and multi-event question performance to measure whether models can retrieve all related episodes after learning them individually. Second, while the original benchmark used multi-paragraph chapters for long-context evaluation, we instead generate simpler single-sentence events, which allows to better isolate the binding challenge from potential side effects arising from, e.g., context length limitation.

### 2.1 EPISODIC WORLD MODEL AND SYNTHETIC NARRATIVE GENERATION

Following Huet et al. (2025), each event in our narratives encodes a tuple $(t_i, s_i, ent_i, c_i)$ where $t_i$ denotes time (e.g., "March 5th"), $s_i$ denotes space (e.g., "Tokyo"), $ent_i$ denotes the entity involved (e.g., "Mary"), and $c_i$ denotes the content or action (e.g., "painting"). In our adaptation, each event becomes a single concise sentence rather than a paragraph, allowing us to focus on parametric binding rather than context comprehension. The Appendix illustrates specific examples in Fig 4– 6.

We generate synthetic narratives of $N$ events ($N \in \{10, 30, 100\}$) using the original controlled multiplicity approach with truncated geometric sampling: this ensures entities, dates, and spaces naturally recur across multiple events, creating authentic binding challenges (e.g. "Mary" might appear in 1, 3, or 6+ events distributed across different times and locations). We also enforce uniqueness constraints (no duplicate $t \times s$ pairs) and use canonical representations for unambiguous evaluation. We also adopt their three narrative universes (everyday events in New York, imaginary world news, and sci-fi events) preserving event structure. This synthetic generation provides perfect ground truth for any episodic question, enabling precise evaluation without real-world ambiguity.

### 2.2 QUESTION TYPES AND EVALUATION FRAMEWORK

The episodic memory benchmark models episodic recall as cue-based retrieval, where partial event information (a cue) triggers memory recall. A cue is any combination of elements from the event tuple $(t_i, s_i, ent_i, c_i)$, with asterisks denoting wildcards. For example, the cue $(*, s, *, *)$ asks for all events at location $s$, while $(*, *, ent, *)$ asks for all events involving entity $ent$. Table 1 shows how different cue patterns create different retrieval challenges:

To isolate the binding challenge, we distinguish two fundamental question types. **Single-event questions (SEQ)**, detailed in Tab. 3, have answers found in exactly one event: e.g."Where was Mary on March 5th?" has a unique answer since no two events share the same date. **Multi-event questions (MEQ)**, detailed in Tab 4, require retrieving all events matching a cue: e.g. "List all times

Table 1: Cue patterns and their binding requirements. Patterns with unique answers require no binding; patterns with multiple matches test episodic binding (Examples in Fig. 7–2)

| Cue Pattern | Example Question | Binding Requirement |
|---|---|---|
| $(*, s, ent, c)$ | "What day did Mary paint in Tokyo?" | None (unique answer) |
| $(*, *, ent, *)$ | "List all Mary's activities" | Entity binding (multiple events) |
| $(*, s, *, *)$ | "What happened in Tokyo?" | Location binding (multiple events) |
| $(*, *, ent, c)$ | "When did Mary paint?" | Entity-action binding (multiple events) |

Mary was in Tokyo" requires finding every event where Mary appears in Tokyo. While SEQs test basic memorization, MEQs test whether models can bind related episodes through shared elements.

We evaluate MEQs through three diagnostic tasks of increasing difficulty. **Multi-hit retrieval** asks models to retrieve all matching events given a cue, directly testing exhaustive set retrieval. Models must activate all relevant episodic traces, not just the most salient. **Latest state tracking** requires identifying only the most recent event for a given entity, which still requires binding to compare temporal information across multiple episodes. **Chronological ordering** demands retrieving all events for an entity in temporal order, the most challenging task as it requires both complete binding and temporal structure preservation. These tasks form a hierarchy a models that fail at basic multi-hit retrieval cannot succeed at chronological ordering.

**Evaluation** We use lenient recall: $\frac{|\hat{Y} \cap Y|}{|Y|}$ where $Y$ is the ground truth set and $\hat{Y}$ is the model's prediction that we extract using an LLM as a judge (details in App. 8). We consider an answer correct only if recall equals 1.0 (complete retrieval). Note how we do not penalize hallucinations to isolate the binding challenge effects. We stratify results by ground-truth set size $k \in \{1, 2, 3\text{-}5, 6+\}$ to reveal how performance degrades as more episodes must be bound together.

## 2.3 EPISODIC KNOWLEDGE BINDING DEFINITION

We formally define *episodic knowledge binding* as the ability to retrieve all episodes matching a given cue after learning episodes separately. Given a model $\mathcal{M}$ trained on single-event QA pairs $\{(q_i, a_i)\}_{i=1}^N$ where each $q_i$ queries one aspect of event $E_i$, binding manifests when $\mathcal{M}$ can answer multi-event questions requiring exhaustive retrieval across multiple events sharing common attributes. The binding challenge becomes apparent when models reach high lenient recall on SEQs but fail on MEQs. Crucially, this differs from catastrophic forgetting: binding failure occurs when models remember each fact in isolation but cannot retrieve them together when queried.

## 2.4 SCOPE AND DIFFICULTY OF KNOWLEDGE BINDING

In this work, we focus on knowledge binding within the parametric space of the model (cfr related work in Sec.6 and Appendix: D) in static and continual learning settings.

**Scope: Parametric Memory Integration.** This work investigates whether neural networks can learn to bind and retrieve related episodes through their *parameters*, a question distinct from retrieval-augmented generation (RAG), which stores episodes in external databases. While RAG provides practical utility, it does not address whether gradient-based learning can produce compositional knowledge integration. Understanding parametric binding has implications beyond continual learning: it informs whether current architectures can achieve human-like relational reasoning, how foundation model pretraining captures implicit episodic connections across documents, and what architectural or objective innovations might be necessary. We therefore focus on parametric approaches, leaving external memory systems to complementary research.

**The impracticality of exhaustive multi-event supervision.** We note that programmatically generating comprehensive multi-event training data (MEQAs) is computationally costly across all training paradigms. In static training with large corpora, identifying and linking all related episodes across millions of documents would require sophisticated entity resolution and co-reference systems to handle the complex, multifaceted nature of real-world episodic connections (far beyond our

simplified $(t, s, ent, c)$ episodic model). Similarly in continual learning, maintaining a complete database of past episodes, programmatically enumerating all possible, e.g. entity-location-time, combinations, and dynamically updating all multi-event answers as new events arrive becomes expensive as episodes unfold. The issue might also arise in foundation model pre-training, where episodic connections naturally occur across documents, but the sparse and implicit nature of these connections fails to induce robust binding. This fundamental impracticality, whether in static or continual settings, motivates our search for methods that can achieve binding through more efficient training strategies that do not require exhaustive multi-event supervision – that we therefore consider only for reference.

## 3 Effects of Scaling on Episodic Knowledge Binding

We investigate how the episodic binding challenge manifests across model and narrative scales. Our experiments isolate the binding failure from other confounds like catastrophic forgetting by using static training paradigms, as reference alongside our target continual learning ones.

### 3.1 Experimental Protocol

**Models.** We evaluate Llama models (3B, 8B, 13B parameters) and GPT-4.1 variants to capture behavior across a 10x parameter range, from smaller models with limited memory to larger models with enhanced parametric storage. Hyperparameters are selected via grid search on the Continual-NoReplay baseline (details in Appendix A.4)

**Static training paradigms.** To illustrate the problem we compare two ideal strategies as benchmark, isolating different aspects of binding: (i) **Train(SEQ)**: One-shot fine-tuning on all single-event QA pairs pooled together. By removing sequential interference and temporal confounds, this isolates the pure binding challenge: can models generalize from learning each event separately to inferring all times Mary was in Tokyo? (ii) **Train(SEQ+MEQ)**: One-shot fine-tuning including both single and multi-event QAs. This provides direct supervision for binding and serves as an upper bound for what models can achieve with oracle supervision.

**Narrative scales.** We vary narrative length from 10 to 100 events, testing how binding complexity affects performance as the number of episodes and potential connections grows.

### 3.2 The Binding Gap and Its Scaling Effects

Fig. 1 reveals a universal binding failure across model scales. When trained on single-event questions alone Train(SEQA), all models achieve near-perfect lenient recall (86-96%) on single-event retrieval (SEQ) but catastrophically fail at multi-event retrieval (MEQ). This failure occurs despite static training that eliminates temporal confounds and catastrophic forgetting.

**Partial success at small scale.** At 10 events, models show very limited binding capability, achieving 23-37% MEQ lenient recall – which shows that minimal binding can emerges naturally.

**Sharp degradation with narrative length.** As narratives grow from 10 to 30 events, MEQ performance collapses: the 8B model drops from 34% to 27%, while 3B and 13B models fall to near-zero (4% and 2%). At 100 events, all models converge to 5-17% MEQ lenient recall. This sharp degradation reveals how binding complexity overwhelms the models' parametric indices.

**Model scaling offers no relief.** Across a 10x parameter range (3B to 13B), we observe no consistent improvement in binding capability. The 8B model shows marginal gains over 3B and 13B variants, but this inconsistency suggests architectural limitations rather than capacity constraints. Model scaling, which typically improves knowledge-intensive tasks, fails to address episodic binding.

**Multi-event supervision proves binding is possible with current model capacity.** When provided with direct supervision (Train(SEQA+MEQA)), models achieve 88-100% MEQ lenient recall across all narrative lengths, demonstrating that models can memorize multi event questions with appropriate training signals. However, generating such exhaustive multi-event labels requires maintaining a complete episodic database and enumerating all possible entity-location combinations, computationally intractable for continual learning.

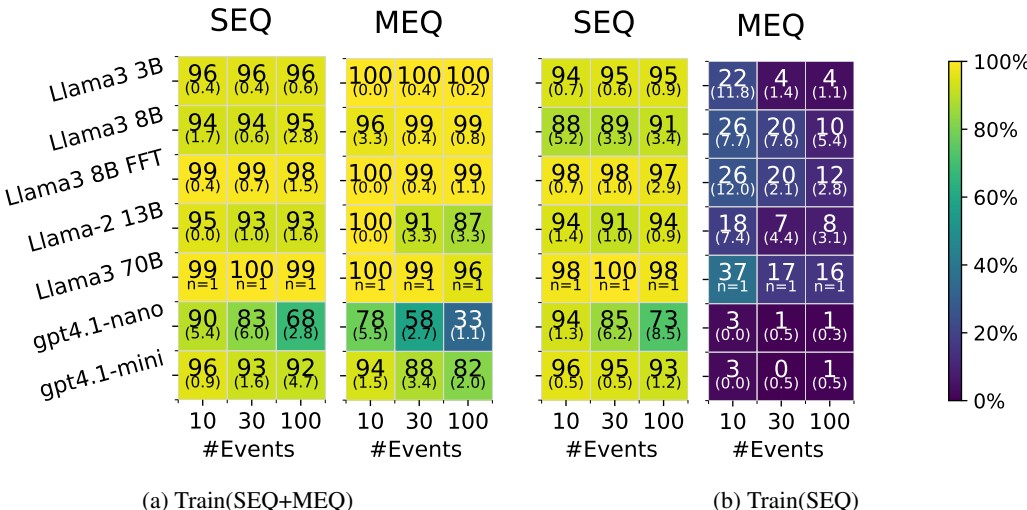

(a) Train(SEQ+MEQ)                                   (b) Train(SEQ)

Figure 1: Average multi-hit performance using lenient recall 2.2 across three narrative types: everyday events in New York, imaginary world news, and sci-fi events. Each cell shows mean lenient recall across 3 runs with standard deviation in parentheses. When n=1, the experiment was conducted once on default New York setting without standard deviation.

**Cross-model consistency validates the challenge.** For the GPT-4.1 family, we conducted experiments using fewer training epochs due to budget constraints. This streamlined evaluation served to confirm that the binding problem generalizes across different model families and architecture — rather than being intented as a direct performance comparison between Llama3 and GPT models.

**The binding problem is not a LoRA artifact.** To verify that our findings are not due to LoRA-specific limitations or suboptimal configurations, we conducted full fine-tuning experiments with Llama3-8B (FFT row). While full fine-tuning yields slightly better results as expected, the improvement remains insufficient to solve the binding problem, confirming that the challenge is fundamental rather than an artifact of parameter-efficient training methods.

**The binding challenge transcends training duration.** Fig. 9 tracks model performance across three metrics as training extends to 3000 epochs. Alongside SEQ and MEQ, we include Commonsense, a dataset of simple general knowledge questions, to check that the model maintains its basic capabilities. The results show that even with very long training, MEQ performance stays low while SEQ accuracy becomes nearly perfect. At 3000 epochs, Commonsense performance drops, indicating the model is memorizing answers rather than learning real binding. This dissociation confirms that episodic binding failure is not a matter of insufficient training iterations, but rather reflects fundamental architectural limitations in handling multi-event binding.

These results establish episodic binding as a fundamental challenge distinct from catastrophic forgetting, one that current architectures cannot overcome through scale alone.

## 4 GENERATIVE CUED REPLAY FOR EPISODIC BINDING

While model scaling fails to address the binding gap, we propose Generative Cued Replay (GCR), inspired by hippocampal memory replay and consolidation. The key insight is that instead of storing past episodes or exhaustively enumerating multi-event combinations, which is infeasible in practice, we can synthesize multi-event training data on-the-fly by querying the model's own parametric memory when learning new events. Note that this approach transforms a classic training problem into a continual learning one, since the synthesized MEQs depend on the current training sample.

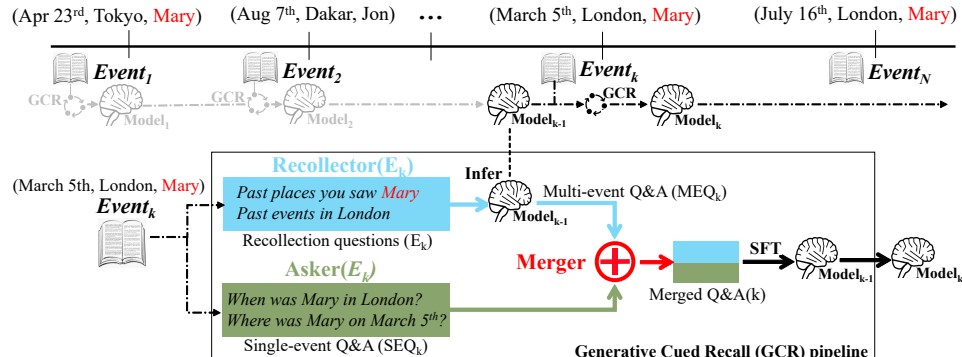

Figure 2: **Generative Cued Replay (GCR) pipeline.** When a new event $E_k$ arrives, the **Recollector** queries the current model $\mathcal{M}_{k-1}$ for related past episodes. The **Asker** generates single-event Q&As about $E_k$. The **Merger** combines recalled episodes with the current event to synthesize multi-event Q&As (e.g., "List all times Mary was in London" → "March 5th, July 16th"), helping the model not just learning the new fact but its connections to the existing episodic memory network.

### 4.1 METHOD OVERVIEW

Figure 2 illustrates the GCR pipeline. When event $E_k$ arrives (e.g., "Mary was in London on March 5th"), the system does not only train on this isolated fact. Instead, the **Recollector** queries the current model $\mathcal{M}_{k-1}$ for related past episodes (other times Mary appeared or other events in Tokyo). Simultaneously, the **Asker** generates single-event questions about $E_k$. The **Merger** then combines recalled episodes with the current event to synthesize multi-event Q&As. This merged training data updates the model via supervised fine-tuning, teaching it not just the new fact but its connections to the existing episodic network. Unlike traditional replay methods that require storing past episodes, GCR leverages the model's own parametric memory as both storage and retrieval mechanism. The GCR pipeline consists of four core components:

**Single-event Asker.** Given a single event's textual description, it generates questions and answers for fine-tuning a learner to recall the event details 12. To make the process fully automated, we use a frontier LLM to synthetically generate finetuning questions starting solely from a textual description of the event. As a control (see Continual-GT below), we also use templated questions based on our ground truth event structure. 6

**Recollector of Related-events.** Retrieves prior episodes *that match the current event* from the learner's parametric memory. We evaluate three retrieval strategies (details Tab. 5): (i) **GCR-Simple**: Uses one basic question per cue type (e.g., "List all events you've seen about Mary", "List all events at location X"; templates in Fig. 9). This tests whether minimal retrieval prompting can trigger episodic binding. (ii) **GCR-Rich**: Uses multiple detailed templated questions per cue, providing richer retrieval context to help the model recall related episodes more comprehensively (details in Fig. 10). (iii) **GCR-Generated**: Instead of fixed templates, employs a frontier LLM to generate diverse, natural recall queries tailored to each event. This tests whether more sophisticated retrieval prompting improves binding performance (details in Fig 11).

**Merger.** Combines recalled episodes with the current event to create synthetic multi-event training questions and answers 13. For example, if the current event is "Mary visited Tokyo in September" and recalled episodes include "Mary visited Tokyo in March", the merger creates questions such as "List all times Mary was in Tokyo" with answer "March, September".

**Hallucination filter.** Since smaller models (3B-13B) often hallucinate during recall, we optionally filter retrieved episodes before merging. The filter removes fabricated entities and details by comparing recollection answers against the ground truth corpus, so that we can disentangle episodic binding challenges from hallucinations 14.

Finally, we use two continual learning baseline. One can realistically obtained using the Asker, the other uses the groundtruh which we use as control: (i) **Continual-Gen**. sequential fine-tuning event-by-event on single-event questions only (generated by the Asker), without any replay mechanism. This baseline faces both catastrophic forgetting and binding challenges, allowing us to measure GCR's improvement. (ii) **Continual-GT.** The same sequential learning pipeline but using groundtruth SEQs instead of Asker-generated ones.

# 5 EVALUATION OF GENERATIVE CUED RECALL

## 5.1 GCR IMPROVES BINDING DESPITE LIMITED RECOLLECTION CAPACITY

Figure 3 evaluates fully automated approaches that require no ground-truth supervision, comparing our GCR method against Continual-Gen, the baseline using LLM-generated questions without any replay mechanism. On 30-event narratives with LlaMA 8B, Continual-Gen achieves only 11% lenient recall for single-match questions and completely fails (0%) when multiple events must be retrieved. GCR doubles single-match performance to 25% and maintains some capability (8-10%) for 2-5 match questions where the baseline fails entirely.

Critically, these results reflect the constraints of testing with LlaMA 8B, a model with limited parametric memory and recollection abilities. When we filter hallucinations (an orthogonal problem to binding) GCR-filtered reveals the true binding improvement: 50% lenient recall for single-match questions, 31% for two-match, and sustained performance even at high multiplicities (19% for 3-5 matches, 6% for 6+ matches). This filtering isolates the binding mechanism from the noise of hallucinated recalls, demonstrating that GCR genuinely improves episodic integration even with a memory-constrained model such as LlaMA 8B. We expect larger gains with models that possess better parametric storage and lower hallucination rates.

The degradation at higher multiplicities is expected: as events accumulate without proper integration, recall errors compound. Each failed recall prevents the model from building complete episodic indices, creating cascading failures for complex multi-event queries. This highlights that binding is fundamentally about information integration during learning, not just retrieval.

## 5.2 COMPARING RETRIEVAL STRATEGIES AND GROUND-TRUTH CONTROLS

Figure 4 provides controlled ablations to understand GCR's mechanisms. The left panel (filtered hallucinations) isolates the binding problem from retrieval noise, while the right panel shows realistic performance including hallucination effects. Two key patterns emerge:

First, the recall-and-merge mechanism drives the improvement, not question quality alone. GCR-Gen filtered outperforms both Continual-Gen (11.1% SEQ) and Continual-GT (17.5% SEQ), despite the latter using ground-truth questions. This confirms that synthetic multi-event training generated through parametric recall is effective.

Second, retrieval strategy matters but not as expected. Simple templated questions (GCR-Simple: 41.3% SEQ filtered) outperform rich templates (GCR-Rich: 32.5%), suggesting that overly specific retrieval prompts may constrain associative recall, consistent with hippocampal theories where partial cues trigger broader pattern completion than detailed ones. Generated queries (GCR-Gen: 50% SEQ filtered) perform best, likely because diverse, natural prompts better activate the model's parametric memory.

Third, we compare against O-LoRa, a contemporary approach for mitigating catastrophic forgetting in continual learning (Chen et al., 2024), that consistently outperformed alternatives (Nayak et al., 2025). While O-LoRa improves single-event performance, it shows minimal gains on multi-event questions, revealing that orthogonal low-rank adaptation preserves general knowledge but fails to address episodic binding. Details are in Appendix A.9.

These results demonstrate that biological-inspired replay mechanisms can already address the binding gap we identify, even with memory-limited models. While exhaustive retrieval remains challenging for high-multiplicity events, GCR provides a foundation for improving episodic binding in parametric continual learning.

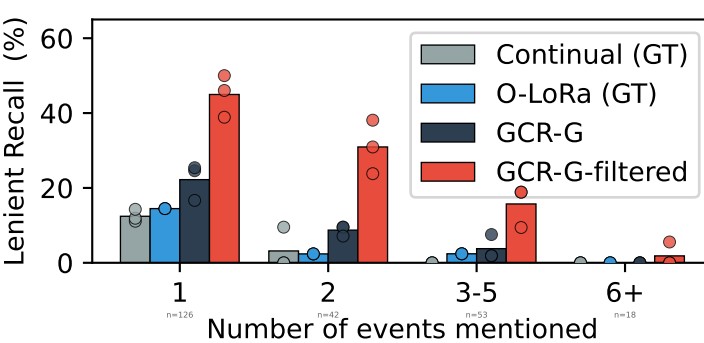

Figure 3: Lenient recall of "fully-automated" versions as a function of the number of events matching a question (30 event narrative, Llama 8B). The difference is statistically significant, as demonstrated via Critical distance (CD) plots deferred to the Appendix C in Fig.10, 11, 12

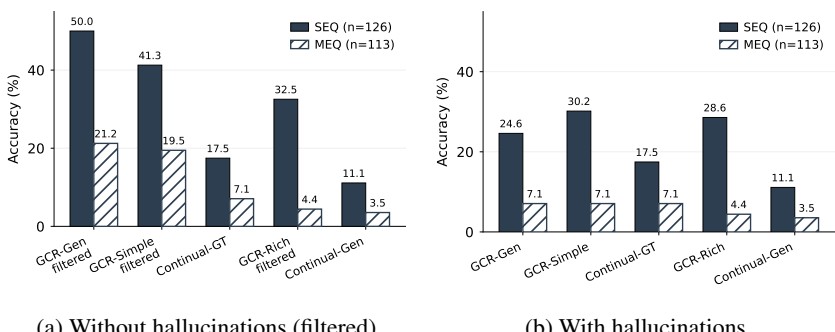

(a) Without hallucinations (filtered)   (b) With hallucinations

Figure 4: Performance of different GCR alternatives/ablations

## 6   RELATED WORK

**Episodic Knowledge Binding: A New Challenge for LLMs**   The ability to update the knowledge of LLMs has revealed a fundamental limitation beyond catastrophic forgetting: models struggle to flexibly access and integrate newly learned information. Recent work demonstrates that LMs fine-tuned on statements like "B's mother is A" fail to answer "Who is A's son?" (Berglund et al., 2023), and more generally cannot perform simple logical deductions or reversals on newly acquired facts (Allen-Zhu & Li, 2024). Yet paradoxically, the same models can compose facts learned separately, inferring a city's identity from distances to Rome and London learned in different LoRA modules (Treutlein et al., 2024). We identify a related but distinct challenge in continual learning: models struggle to bind separately learned episodes that share common elements. This episodic binding failure occurs both in static training scenarios (Figure 1) and in continual learning settings (Figure 4), representing a challenge that, like these generalization failures, goes beyond simple memory retention and can hinder the reasoning capabilities of LMs.

**From Task Retention to Knowledge Integration**   Traditional continual learning research has focused primarily on preventing catastrophic forgetting through techniques like Elastic Weight Consolidation (Kirkpatrick et al., 2017), experience replay (Rebuffi et al., 2017b), and architectural isolation (Rusu et al., 2016). While these classic methods help preserve previously learned information and can mitigate our problem, they were designed for an era before LLMs when the primary goal was learning sequences of classification tasks or adapting to new domains, not continuously integrating declarative knowledge. For instance, classic methods ensured that a model trained on MNIST could still perform well after learning CIFAR-10, or that a sentiment classifier trained on movie reviews could adapt to product without forgetting. With LLMs, however, the challenge shifts fundamentally: models must not only retain facts learned at different times but also perform flexible reasoning and generalization across them. While more recent approaches like O-LoRA (Chen et al., 2024) adapt parameter-efficient fine-tuning to continual learning by enforcing orthogonality constraints on LoRA updates to prevent task interference, which we adapt to our episodic setting (Appendix A.9). The episodic binding challenge emerges as a new challenge in this context because LLMs are expected to connect related episodes through shared elements. This capability goes

beyond simple task retention and requires maintaining accessible relationships between separately learned facts, and has no trivial counterpart in classic task-incremental continual learning.

**Generative Cued Replay: Beyond Exemplar Selection**  In continual learning with LLMs, a critical question emerges: which data should be replayed? The literature offers rich strategies developed mainly for task or class-incremental scenarios: exemplar replay with uniform sampling (Rebuffi et al., 2017a), gradient-constrained methods like Gradient Episodic Memory (Lopez-Paz & Ranzato, 2017a) and A-GEM (Chaudhry et al., 2019), meta-learning approaches such as Meta-Experience Replay (Riemer et al., 2018) that shape replay through optimization objectives, and targeted sampling schemes like Maximally Interfered Retrieval (Aljundi et al., 2019). More recently, gradient-based coreset selection methods refine replay by selecting past examples whose weighted gradients best approximate the full training history (Tiwari et al., 2022), while Huang et al. (2024) move toward self-synthesized rehearsal data, though still missing the factual knowledge connection. Recent work highlights how dataset bias transfers across tasks in continual learning, proposing Balanced Greedy Sampling to mitigate this through balanced exemplar selection (Lee et al., 2023). These efforts influenced our design, yet GCR departs in two key ways. First, GCR focuses on integrating facts within a task rather than adding new labels. Second, instead of selecting exemplars via external heuristics, GCR generates queries that probe the model's parametric knowledge.

**Fact Integration in Static vs Continual Settings**  While our work addresses episodic binding in continual learning, parallel efforts have examined similar integration challenges in static training contexts. Lampinen et al. (2025a) demonstrate that LLMs struggle with flexible factual generalization from fine-tuning models trained on "B's mother is A" fail to answer "Who is A's son?" but handle these reversals easily when the same facts appear in-context. Their controlled experiments reveal a fundamental asymmetry: in-context learning generalizes better to reversals, syllogisms, and multi-hop inferences than gradient-based fine-tuning on the same data. Similarly, Lampinen et al. (2025b) show that models fail to reuse implicitly learned relational facts, requiring oracle retrieval and specific training to enable flexible fact usage. While Yang et al. (2025) propose EntiGraph to generate synthetic data from knowledge graphs, their approach ultimately relies on retrieval mechanisms rather than parametric integration. These findings clarify why current systems fail at flexible factual integration but focus on static training scenarios with full datasets available upfront. In contrast, our episodic binding problem instance specifically targets continual learning, where facts arrive sequentially and must be bound incrementally without access to the full training history.

We defer a comprehensive review of the relevant literature to Appendix D.

## 7 CONCLUSIONS

This work reveals a fundamental limitation in how LLMs encode and retrieve episodic knowledge: models trained on individual episodes fail to spontaneously bind them through shared elements. This effect persists across model scales(3B-13B) and even in GPT-4.1, suggesting it reflects architectural and objective limitations rather than capacity constraints.

Although we later studied the problem in the continual learning setting which fits better our rehersal-based solution, the binding problem emerges in static training too, where catastrophic forgetting plays no role. Our static results suggest that current training creates lookup tables rather than queryable episodic indices. Faced with the impracticality of generating exhaustive multi-event supervision (which would require perfect knowledge of all episodic connections across massive corpora) we proposed Generative Cued Replay. GCR leverages the model's own parametric memory to synthesize multi-event training data on-demand. While our implementation shows promising improvements, it also reveals multiple opportunities for advancement along all the components of our approach: the *Asker*, the *Recollector* as well as the *Merger* could benefit from more sophisticated synthesis.

Beyond improving these components, fundamental questions remain unexplored. For example, how does episodic binding manifest in foundation model pretraining, where episodes naturally span documents but connections remain implicit? Does the sparse, distributed nature of episodic links in real text manifest itself in frontier model knowledge? Could alternative training objectives, perhaps explicitly encouraging multi-event integration, help models develop the necessary inductive biases?

# 8 REPRODUCIBILITY STATEMENT

Details about narrative generation and LLM judge prompts are available in Appendix. Anonymized code is available Anonymous (2025)

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

## A  IMPLEMENTATION DETAILS

### A.1  EVENTS GENERATION

We construct possible universes from atomic components comprising Huet et al. (2025): temporal boundaries (start/end dates), 100 distinct first names, last names, locations, event contents, and 30 unique details per content type. Different component sets are used depending on the narrative style. The static universe defines $N_{\text{universe}} = 100$ dates (sampled from the temporal range), full names (randomly combining first and last names), locations, and contents (shuffled from raw materials), while content details remain unchanged. All items are ensured to be unique, with a seed parameter for reproducibility. For simplicity, Listing 1 presents one representative component set.

```
# temporal
start_date = datetime(2024, 1, 1)
end_date = datetime(2026, 12, 31)

# entities
first_names = ['Henry', 'Evelyn', 'Alexander',...]
last_names = ['Hernandez', 'Lopez', 'Gonzalez',...]

# locations
locations = ['Central Park', 'Times Square', 'Brooklyn Bridge',...]

# contents
contents = ['Educational Workshop', 'Yoga Retreat', 'Photography
    Exhibition',...]

# contents details
content_details = {
  'Educational Workshop': ['Performed musical number', 'Discussed costume
      design','Explained method acting techniques',...],
  'Yoga Retreat': ['Led meditation session', 'Demonstrated breathing
      techniques','Guided mindfulness exercises',...],
  'Photography Exhibition': ['Unveiled new collection', 'Explained
      composition techniques','Discussed lighting methods',...]
}
```

Listing 1: Excerpt of the raw materials with default universe style

The atomic events are structured as tuples containing five elements: temporal information, spatial location, entity identity, event type, and specific action details. Listing 2 illustrates the event structure with color-coded components, where each tuple provides the complete metadata required for narrative generation while maintaining clear semantic boundaries between different information types.

```
events[0] = ['June 14, 2025', 'Washington Square Park', 'Mila Gonzalez',
    'Theater Performance', 'Explained method acting techniques'],
events[1] = ['February 27, 2026', 'Washington Square Park', 'Henry Reed',
    'Fashion Show','Revealed future collections'],
events[2] = ['June 14, 2025', 'Statue of Liberty', 'Brooklyn Ross', '
    Fashion Show','Explained fabric choices'],
events[3] = ['November 13, 2026', 'One World Trade Center', 'Levi
    Rodriguez', 'Theater Performance','Discussed theater technology']
```

Listing 2: Excerpt of the first four events with entity highlighting. Dates are in @4brown@4, locations in @1green@1, persons in @2blue@2, and events in @3magenta@3.

#### A.1.1  NARRATIVE GENERATION PROMPTS

Each event is transformed into a narrative paragraph using the template prompt shown in Listing 3. The template incorporates placeholders for event metadata (date, location, entity, content, and specific details) and style parameters, which are populated from the static universe components before

being processed by the language model. The prompt enforces strict constraints on narrative length (20-30 words), temporal scope (single day), and spatial boundaries (single location) to ensure consistency across generated paragraphs. Additionally, the template mandates verbatim inclusion of all key information elements, full names, dates, locations, and event details, to maintain factual accuracy while allowing stylistic variation within the specified narrative framework

```
Write a brief and short text, do not use more than 20-30 words, excerpt
    in a style style about entity attending a content. Please be ultra
    brief don't use more than 20/30 words.

The story takes place on date, at location, where entity
    content_single_detail. Follow these guidelines:

Structure and Information Reveal:
str_numbering, keep in mind this is a short story text.

2. Reveal key information:
- Full location 'location': must appear verbatim
- Full date 'date': must appear verbatim
- Full name 'entity': must appear verbatim
- Full detail that 'first_name content_single_detail': must appear
    verbatim

Content and Setting:
1. Include the detail that first_name content_single_detail.
2. Limit the timeframe to a single day and confine all action to
    location. Limit also the number of words, use a little number!

Characters:
1. Omit background information about first_name and other characters.

Style and Tone:
1. Incorporate elements of the style style, including style_description
    .
2. Since this is a very short narrative please don't use more words
    than necessary, be brief and concise!

Restrictions:
1. Only mention location and date; avoid other locations or dates.
2. Exclude explicit introductions, conclusions, or character
    backgrounds.
3. Focus exclusively on the events of this particular content.
4. Do not use a too common starting sentence.
5. Do not use more than 20-30 words for each paragraph

Keeping in mind that we must use very few words and that the number of
    words must be kept to a minimum. up to 3 sentences!
```

Listing 3: Prompt template for short narrative generation with variable placeholders. The highlighted elements are replaced by the event and event meta-data values.

Examples of narratives with 10 events

We generate episodic memory benchmarks of varying scales by creating narratives with 10, 30, 100 events for each universe configuration. To assess potential scale effects and ensure statistical robustness, this generation process is repeated across three distinct static universes 4 5 6. This systematic approach yields a comprehensive evaluation dataset spanning multiple narrative lengths and universe instantiations, enabling assessment of how episodic memory performance varies with temporal complexity and contextual scale. For brevity, we present only the 10-event narratives from each of the three universes.

```
Chapter 1 On June 14, 2025, at Washington Square Park, Mila Gonzalez
    captivated a small crowd. She eloquently explained method acting
```

```
techniques, demonstrating with impromptu performances. The audience
    watched, mesmerized by her expertise.

Chapter 2 At Washington Square Park on February 27, 2026, Henry Reed
    unveiled his groundbreaking designs. Models strutted, showcasing
    futuristic attire. Henry revealed future collections, leaving the
    audience in awe.

Chapter 3 At the Statue of Liberty on June 14, 2025, Brooklyn Ross
    captivated the audience. Amidst the iconic backdrop, she explained
    fabric choices with precision. The fashion show attendees hung on her
     every word.

Chapter 4 At One World Trade Center on November 13, 2026, Levi Rodriguez
    attended a theater performance. During intermission, he discussed
    theater technology with fellow attendees, marveling at the venue's
    cutting-edge systems.

Chapter 5 On May 11, 2026, at Washington Square Park, Levi Rodriguez
    explained method acting techniques to a captivated audience. His
    impromptu demonstration drew curious onlookers, transforming the park
     into an unexpected theater classroom.

Chapter 6 At High Line on February 27, 2026, Samuel Parker attended a
    tech hackathon. Amid the buzz of innovation, he discussed agile
    methodologies with fellow participants. The event sparked new ideas
    and collaborations.

Chapter 7 At the Metropolitan Museum of Art on February 27, 2026, Levi
    Rodriguez attended an educational workshop. Surrounded by ancient
    artifacts, he discussed career implications with fellow participants.
     The setting inspired thoughtful dialogue about professional futures.

Chapter 8 At the Metropolitan Museum of Art on May 11, 2026, Scarlett
    Thomas explained choreography during a captivating fashion show.
    Models gracefully showcased avant-garde designs as Scarlett's
    instructions guided their movements.

Chapter 9 At the Metropolitan Museum of Art on September 22, 2026, Carter
     Stewart attended a fashion show. Amidst the glittering runway,
    Carter discussed music selection, his input shaping the event's
    ambiance.

Chapter 10 On June 14, 2025, Henry Reed participated in brainstorming at
    Metropolitan Museum of Art. The educational workshop buzzed with
    creative energy as attendees explored innovative exhibition concepts.
```

Listing 4: **Style: Default**

```
Chapter 1 On June 14, 2025, Mila Gonzalez witnessed a bridge collapse in
    Luzon Region. The structure buckled without warning, plunging
    vehicles into the river below. Screams pierced the air as Mila
    watched, frozen in disbelief.

Chapter 2 On February 27, 2026, Henry Reed witnessed a residential tower
    fire in Luzon Region. Smoke billowed as flames engulfed the structure
    . Sirens wailed through the night air.

Chapter 3 On June 14, 2025, Brooklyn Ross witnessed a residential tower
    fire in North Dakota. Flames engulfed the building, casting an eerie
    glow across the night sky. Sirens wailed as firefighters battled the
    inferno.

Chapter 4 On November 13, 2026, Levi Rodriguez witnessed a bridge
    collapse in Rajasthan. The structure crumbled before his eyes,
```

```
sending debris into the river below. Chaos ensued as emergency
services rushed to the scene.

Chapter 5 On May 11, 2026, Levi Rodriguez witnessed a bridge collapse in
    Luzon Region. The structure buckled without warning, plunging
    vehicles into the river below. Chaos erupted as emergency services
    rushed to the scene.

Chapter 6 On February 27, 2026, Samuel Parker witnessed a volcanic
    eruption in Greater Mumbai. The sky darkened as ash billowed,
    blanketing the city. Chaos erupted as residents fled the
    unprecedented disaster.

Chapter 7 On February 27, 2026, Levi Rodriguez witnessed a flash flood
    emergency in Luang Prabang Province. Torrential rains unleashed a
    deluge, transforming streets into rivers. Residents scrambled for
    safety as water levels rose rapidly.

Chapter 8 On May 11, 2026, Scarlett Thomas witnessed a residential tower
    fire in Luang Prabang Province. Flames engulfed the building as
    sirens wailed. Residents fled while firefighters battled the inferno.

Chapter 9 On September 22, 2026, Carter Stewart witnessed a residential
    tower fire in Luang Prabang Province. Flames engulfed the building,
    casting an eerie glow over the night sky. Sirens wailed as emergency
    responders rushed to the scene.

Chapter 10 On June 14, 2025, Henry Reed witnessed a flash flood emergency
     in Luang Prabang Province. Torrential rain unleashed chaos, sweeping
     away vehicles and forcing evacuations. Henry watched helplessly as
     the surging waters transformed familiar streets into raging rivers.
```

Listing 5: **Style: News**

```
Chapter 1 On June 14, 2225, at Europa Subsurface Laboratory, Mila
    Gonzalez witnessed a antimatter cascade. Particles collided,
    releasing a blinding flash. The event shook the facility, leaving
    Mila awestruck.

Chapter 2 At Europa Subsurface Laboratory on February 27, 2226, Henry
    Reed witnessed a fusion core breach. Alarms blared as containment
    fields failed. Blinding light engulfed the chamber, leaving Henry
    stunned and breathless.

Chapter 3 On June 14, 2225, at Mars Valles Industrial Hub, Brooklyn Ross
    witnessed a fusion core breach. Alarms blared as blinding light
    erupted. Technicians scrambled, their faces etched with panic.

Chapter 4 On November 13, 2226, at Luna Shackleton Crater Colony, Levi
    Rodriguez witnessed a antimatter cascade. The event unfolded swiftly,
     bathing the lunar outpost in an eerie glow. Spectators watched in
    awe as energy rippled across the crater's edge.

Chapter 5 At Europa Subsurface Laboratory on May 11, 2226, Levi Rodriguez
     witnessed a antimatter cascade. Particles collided, unleashing a
    brilliant flash. The event left him awestruck, etching the moment in
    scientific history.

Chapter 6 On February 27, 2226, at Mercury Twilight Observatory, Samuel
    Parker witnessed a cryo-pod integrity breach. Alarms blared as frozen
     vapor billowed from the ruptured chamber. Technicians scrambled to
    contain the crisis.

Chapter 7 At Luna Oceanus Trading Post on February 27, 2226, Levi
    Rodriguez witnessed a plasma conduit rupture. Searing blue light
```

```
     flooded the chamber. Alarms blared as technicians scrambled to
     contain the breach.

Chapter 8 On May 11, 2226, Scarlett Thomas witnessed a fusion core breach
     at Luna Oceanus Trading Post. Alarms blared as radiation levels
     spiked. Personnel evacuated, leaving Scarlett to face the unfolding
     catastrophe.

Chapter 9 On September 22, 2226, Carter Stewart witnessed a fusion core
     breach at Luna Oceanus Trading Post. Alarms blared as the facility
     shuddered. Technicians scrambled to contain the erupting plasma,
     their faces etched with panic.

Chapter 10 On June 14, 2225, at Luna Oceanus Trading Post, Henry Reed
     witnessed a plasma conduit rupture. Alarms blared as blue-white
     energy surged. Technicians scrambled to contain the breach, averting
     catastrophe.
```

Listing 6: **Style: Sci-Fi**

Examples of questions By construction, each event in our episodic memory framework is composed of four fundamental dimensions: time (t), space (s), entity (ent), and content (c). This structured representation enables systematic querying across all possible combinations of these dimensions.

Tab.2 shows how the episodic benchmark is built. Here we can see some examples of questions.

Table 2: Episodic memory questions based on cue composition and retrieval types (Taken from Huet et al. (2025) with permission).

| Cue | Description | Retrieved trace (id) | Template question (corresponding to ⋆) |
|---|---|---|---|
| (t, *, *, *) | Events at a specific time | - Spaces (0)
- Entities (1) ⋆
- Contents (2) | ⋆ Consider all events that happened on {t}. Provide a list of all protagonists involved in any of these events, without describing the events themselves. |
| (*, s, ent, *) | Events involving entities at a specific location | - Times (18)
- Contents (19) ⋆ | ⋆ Reflect on {ent}'s experiences at {s}. Describe all the key events they've been involved in at this location, focusing on what happened rather than when it occurred. |
| (*, s, ent, c) | Events with specific location, entities, and content | - Times (27) ⋆ | ⋆ Consider all events involving both {ent} and {c} at {s}. Provide a list of all dates when these events occurred, without describing the events. |
| (t, s, ent, c) | Events with specific time, location, entities, and content | - Full event details (29) ⋆ | ⋆ Provide a comprehensive account of what happened involving {ent} and {c} at {s} on {t}. Include all relevant details about the event(s), including what occurred and any other pertinent information. |
| (*, *, ent, *) | Retrieves the most recent known location of an entity | - Times [latest] (30)
- Spaces [latest] (31) ⋆
- Contents [latest] (32) | ⋆ What is the most recent location where {ent} was observed in the story's chronological timeline? |
| (*, *, ent, *) | Retrieves a chronological list of dates when an entity was observed | - Times [chrono.] (33) ⋆
- Spaces [chrono.] (34)
- Contents [chrono.] (35) | ⋆ Provide a chronological list of all dates when {ent} was observed, from earliest to latest in the story's timeline. |

If the answer is contained in just a single chapter we define the question as SEQ. Otherwise if the answer is in more than 1 we have MEQ.

Table 3: Example of SEQ

| question | cue | cue_completed | ret_type | get | correct_answer | chapters |
|---|---|---|---|---|---|---|
| Reflect on September 22, 2026. Describe all the key events that occurred on this date, focusing on what happened rather than who was involved or where it took place. | (t, *, *, *) | ({September 22, 2026}, *, *, *) | Event contents | all | [Fashion Show] | [9] |
| Consider all events that happened on November 13, 2026. Provide a list of all protagonists involved in any of these events, without describing the events themselves. | (t, *, *, *) | ({November 13, 2026}, *, *, *) | Entities | all | [Levi Rodriguez] | [4] |
| Reflect on September 22, 2026. Provide a list of all protagonists involved in any of these events, without describing the events themselves. | (t, *, *, *) | ({September 22, 2026}, *, *, *) | Entities | all | [Carter Stewart] | [9] |

Table 4: Example of MEQ

| question | cue | cue_completed | ret_type | get | correct_answer | chapters |
|---|---|---|---|---|---|---|
| Reflect on events related to Educational Workshop. Provide a list of all protagonists involved in these events, without describing the events. | (*, *, *, c) | (*, *, *, {Educational Workshop}) | Entities | all | [Henry Reed, Levi Rodriguez] | [10, 7] |
| Consider all events involving Educational Workshop. List all the locations where these events took place, without mentioning the events themselves. | (*, *, *, c) | (*, *, *, {Educational Workshop}) | Spaces | all | [Metropolitan Museum of Art] | [10, 7] |
| Recall all events related to Educational Workshop. Provide a list of all dates when these events occurred, without describing the events. | (*, *, *, c) | (*, *, *, {Educational Workshop}) | Times | all | [February 27, 2026, June 14, 2025] | [10, 7] |

Since all Llama models are instruction-tuned, we add contextual framing to specify that the evaluation involves fictional events and entities. This ensures responses are based on the provided narrative rather than pre-existing knowledge. The following examples show our prompt format for just a couple of questions:

```
[
  {
    "messages": [
      {
        "content": "You are an expert in memory tests regarding the
    fictional book \"Synaptic Echoes 2026: The Neuro-Temporal Paradox of
     Episodic Precognition\".",
        "role": "system"
      },
      {
        "content": "This question is about the book \"Synaptic Echoes
    2026: The Neuro-Temporal Paradox of Episodic Precognition\". All
    events in this book are purely fictional and do not correspond to
    real-world timelines. Please answer based solely on the content of
    this fictional story.\n\n Question: Reflect on all events involving
    Samuel Parker. Provide a list of all dates when these events
    occurred, without describing the events.",
        "role": "user"
      },
```

```
1080          {
1081            "content": "February 27, 2026",
1082            "role": "assistant"
1083          }
1084        ]
1085      },
1086      {
1087        "messages": [
1088          {
1089            "content": "You are an expert in memory tests regarding the
             fictional book \"Synaptic Echoes 2026: The Neuro-Temporal Paradox of
              Episodic Precognition\".",
1090            "role": "system"
1091          },
1092          {
1093            "content": "This question is about the book \"Synaptic Echoes
             2026: The Neuro-Temporal Paradox of Episodic Precognition\". All
             events in this book are purely fictional and do not correspond to
             real-world timelines. Please answer based solely on the content of
             this fictional story.\n\n Question: Consider all events involving
             Henry Reed at Washington Square Park. Provide a list of all dates
             when these events occurred, without describing the events.",
1099            "role": "user"
1100          },
1101          {
1102            "content": "February 27, 2026",
1103            "role": "assistant"
1104          }
           ]
1105      },
1106      {
1107        "messages": [
1108          {
1109            "content": "You are an expert in memory tests regarding the
             fictional book \"Synaptic Echoes 2026: The Neuro-Temporal Paradox of
              Episodic Precognition\".",
1110            "role": "system"
1111          },
1112          {
1113            "content": "This question is about the book \"Synaptic Echoes
             2026: The Neuro-Temporal Paradox of Episodic Precognition\". All
             events in this book are purely fictional and do not correspond to
             real-world timelines. Please answer based solely on the content of
             this fictional story.\n\n Question: Consider the events involving
             Henry Reed on June 14, 2025. List all the locations where these
             events took place, without describing the events themselves.",
1119            "role": "user"
1120          },
1121          {
1122            "content": "Metropolitan Museum of Art",
             "role": "assistant"
1123          }
1124        ]
1125      }
1126    ]
```

Listing 7: Example prompt structures for finetuning q/a

## A.2 EVALUATION PROMPTS

We build on the evaluation prompt from Huet et al. (2025). Our adaptation employs Gemini 2.5 Flash as an automated judge that compares model responses against ground truth answers, assigning continuous scores between 0 and 1 based on accuracy and completeness. This approach enables

scalable evaluation across our comprehensive benchmark while maintaining consistent scoring criteria that account for synonyms, paraphrases, and partial matches, allowing us to systematically measure how well models retrieve specific episodic details at scale. The prompt is shown in 8

```
RESEARCH EVALUATION TASK:
You are conducting an academic evaluation of AI system accuracy. Below
    is factual reference data and an AI-generated response that needs
    objective assessment.

You are an expert judge evaluating the accuracy of an AI-generated
    answer against a known groundtruth. Questions can probe for
    different types or aspects, like what actions or events took place,
    what people were involved, what were the dates, or what were the
    locations or spaces.

Question type: retrieval_type
Groundtruth: correct_answer_long
AI-generated answer: llm_answer

Your task:
- Identify all unique items in the AI-generated answer that are
    relevant to the question type. Answer an empty list [] for this
    field in case of at least one negative information (e.g., when the
    answer begins by telling there is no information, or cannot answer)
- Determine a matching score between 0 and 1 for each ground truth item
    . Give 1 if the item has been found in the relevant items of the AI-
    generated answer, considering synonyms, paraphrases, or close
    meanings. Give 0.5 if the item could be considered related to any AI
    -generated item but without being explicitly stated as such. Give 0
    if the item missed mentioning a specific AI-generated item.
- Provide a brief explanation of the evaluation
adding_text

Provide your evaluation in the following JSON format, no markdown
    formatting or code blocks:
{
    "identified_items_in_AI_answer": ["AI_answer_item_1", "
    AI_answer_item_2", ...],
    "matching_score": json.dumps(d)
    "explanation": "Brief explanation of your evaluation"
}
```

Listing 8: Evaluation prompt template

Building upon the LLM judge evaluation, we define a binary correctness metric that addresses model hallucination. A response is marked as correct only if it contains all ground truth elements, no missing information is tolerated. However, we do not penalize models for providing additional correct information beyond what is required. For example, if the ground truth specifies 3 elements and a model returns 5 elements, the answer is correct if all 3 required elements are present among the 5 provided. This strict recall requirement ensures models must demonstrate complete episodic memory retrieval while allowing for comprehensive responses that exceed minimum requirements.

### A.3 HYPERPARAMETER SELECTION

We perform grid search over learning rates ($10^{-5}$ to $10^{-3}$), batch sizes (4 to 32), and epochs (1 to 60). For computational efficiency, we use the Continual-NoReplay baseline (sequential training without replay) as the optimization target. Selected hyperparameters are then applied consistently across all conditions to ensure fair comparison.

### A.4 LLM AS A JUDGE: REPRODUCIBILITY ANALYSIS

To address concerns about the reproducibility and reliability of LLM-based evaluation, we conducted an inter-rater reliability study comparing three state-of-the-art LLM judges (GPT-5-mini,

Claude-Haiku-4.5, and Gemini-2.5-flash) against human expert annotations on a sample of 100 question-answer pairs. As shown in Figure 5, all three models demonstrate exceptionally high agreement with human judgments (98-99%), indicating that our evaluation methodology is both reliable and reproducible across different LLM implementations. The near-perfect consensus among diverse model architectures and providers suggests that the evaluation task is sufficiently objective and well-defined, mitigating concerns about evaluation variance across different LLM versions or sampling parameters.

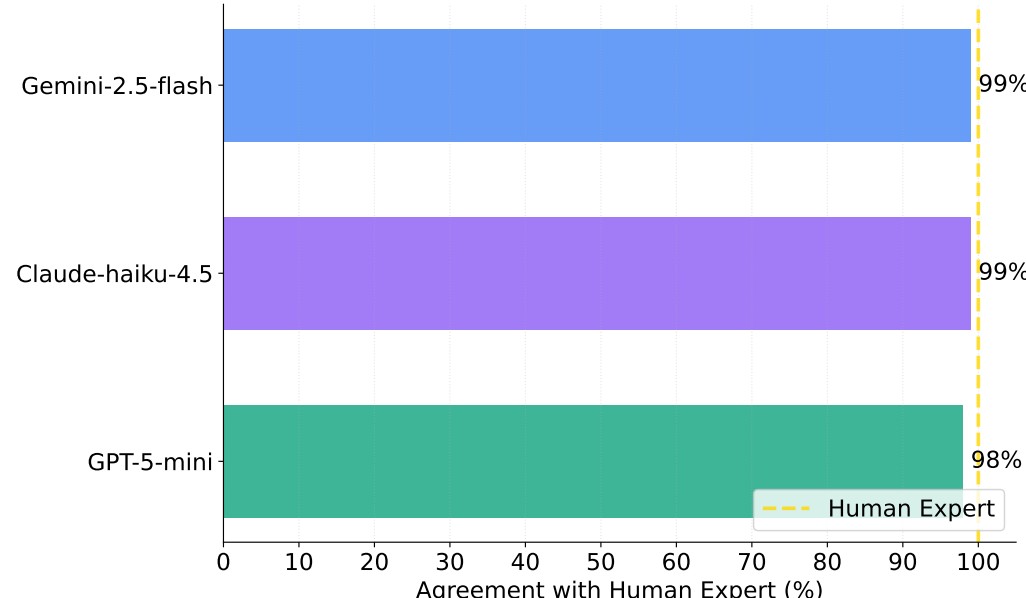

Figure 5: Comparison of 3 llm as a Judge and 1 Human Expert

## A.5 BUILDING GCR COMPONENTS

We introduce Generative Cued Replay (GCR), inspired by hippocampal memory consolidation. GCR generates multi-event question-answer pairs during training using the model's internal knowledge, eliminating the need for explicit episode storage.

### A.5.1 GCR-SIMPLE

GCR-Simple employs four template-based recollection queries that target different aspects of episodic memory retrieval. As shown in Listing 9, these templates prompt the model to recall information based on temporal markers ('t'), spatial locations ('s'), entity identities ('ent'), and content types ('c'). Each template follows a consistent "List everything you remember" structure while systematically probing distinct dimensions of the learned episodes, enabling comprehensive memory consolidation across the narrative's key components.

```
f"List everything you remember about t in the book."
f"List everything you remember in this book about s."
f"List everything you remember in this book about ent."
f"List everything you remember in this book about c."
```

Listing 9: Recollection question templates GCR-Simple

### A.5.2 GCR-RICH

GCR-Rich extends the basic recollection approach with twelve sophisticated template queries that probe multi-dimensional episodic associations. As shown in Listing 10, these templates systematically explore cross-references between temporal, spatial, entity, and content dimensions while maintaining selective focus. Each query follows a structured pattern: given one episodic dimension (e.g., a specific date 't' or location 's'), the model must retrieve information from a different dimension (locations, entities, events, or dates) without describing intermediate details.

```
"Recall all the events that occurred on t. Without describing the
    events,list all the unique locations where these events took place."

"Consider all events that happened on t. Provide a list of all
    protagonists involved in any of these events, without describing the
     events themselves."

"Reflect on t. Describe all the key events that occurred on this date,
    focusing on what happened rather than who was involved or where it
    took place."

"Think about all events that have occurred at s. Provide a list of all
    dates when these events took place, without describing the events."

"Consider the location s. List all protagonists that have been involved
     in any events at this location, without mentioning the events
    themselves."

"Recall the various events that have taken place at s. Describe what
    happened during these events, focusing on the actions or occurrences
     rather than the timing or people involved."

"Reflect on all events involving ent. Provide a list of all dates when
    these events occurred, without describing the events."

"Consider all events that ent has been involved in. List all the
    locations where these events took place, without mentioning the
    events themselves."

"Think about ent's experiences. Describe all the key events they've
    been involved in, focusing on what happened rather than when or
    where it occurred."
```

```
"Recall all events related to c. Provide a list of all dates when these
    events occurred, without describing the events."

"Consider all events involving c. List all the locations where these
    events took place, without mentioning the events themselves."

"Reflect on events related to c. Provide a list of all protagonists
    involved in these events, without describing the events."
```

Listing 10: Twelve template questions for retrieval queries

### A.5.3 GCR-GENERATED

GCR-Generated replaces fixed templates with dynamic question generation using a frontier LLM to create contextually relevant memory probes. As shown in Listing 11, this approach employs a sophisticated prompt that instructs the question generator to identify key entities (people, places, events, objects, relationships) within each narrative excerpt and generate targeted recall queries. The generator creates diverse probe types including entity identification, temporal sequencing, and chronological reconstruction questions, while remaining agnostic to whether entities are appearing for the first or multiple times.

```
You are a question generator seeing a text excerpt for the first time.
    Your job is to generate questions that will help an LLM model (which
     MAY have seen previous chapters of the same document) recall
    potential connections and maintain temporal coherence.

CURRENT EXCERPT:
content_text

DOCUMENT CONTEXT: document_context

TASK: Generate memory-probing questions about entities in this excerpt.
     You DON'T know if these entities appeared before - the questions
    should work whether this is their first appearance or not.

1. **Identify Key Entities** in this excerpt, e.g.:
   - People (names, roles)
   - Places (locations, settings)
   - Events (actions, occurrences)
   - Objects (important items)
   - Relationships (between any entities)

2. **Generate Open-Ended Memory Probes to the model**, e.g.:
   - "What do you remember about [entity] from earlier in this
    narrative, if anything?"
   - "Have you encountered [entity] before in this story? If so, when
    and where?"
   - "Is this your first time seeing [entity] in this narrative?"

3. **Temporal & Sequential Probes**, e.g.:
   - "If you've seen [entity] before, what has changed since then?"
   - "Where does this event fit in the sequence of events you've read
    about?"
   - "What events, if any, led up to this moment, based on what you
    remember?"

4. **Full Sequence Report Questions** (CRITICAL for temporal ordering),
     e.g.:
   - "Can you list ALL the places [entity] has been, in chronological
    order?" (if entity is not a place, obviously)
   - "What is the complete sequence of [entity]'s appearances so far?"
   - "What prior events happened in [entity]" (if entity is a location)
```

```
     - "Trace the timeline: What happened first, then next, leading to
       this point?"
     - "If you've seen these entities before, what order did you
       encounter them in?"
     - "Reconstruct the journey: How did [entity] get from their first
       appearance to here?"
     - "What is the chronological order of major events involving [entity
       ]?" (let it be a place or a person etc..)
     - "If [location] has appeared before, what events occurred there in
       chronological order?"

5. Now that you understand the goal, adapt your questions to the
   CURRENT_EXCERPT content.

6. Keep in mind that the tested model will receive a context prefix
   text followed by your question, make sure your question fits.

OUTPUT FORMAT:
{
  "questions": [
    {
      "entity": "the entity being probed",
      "entity_type": "person/place/object/event/relationship",
      "question": "the memory probe question",
      "probe_type": "identity/location/state/temporal/relationship/
    sequence_report/chronology"
    }
  ]
}

Generate as many questions as needed, but do not exceed 20, pick the
   most significant. Questions should be open-ended enough to work
   whether this is the entity's first or fifth appearance, but specific
    to elicit useful responses. Avoid redundancy.
```

Listing 11: Recollection question templates GCR-Generated

Here we present a comparative analysis of the different question generation approaches using a single chapter corpus, as detailed in Table 5.

### A.6 SINGLE-EVENTS ASKER

Here we present the prompt template used to generate question-answer pairs from individual narrative chapters for fine-tuning purposes. As shown in Listing 12, the prompt integrates the complete chapter text (`data_tuple[1]`) along with the five key factual elements: date (`data_tuple[0][0]`), location (`data_tuple[0][1]`), entity (`data_tuple[0][2]`), event type (`data_tuple[0][3]`), and specific detail (`data_tuple[0][4]`).

```
You are tasked with creating 15 high-quality question-answer pairs from
    the provided text to help fine-tune a language model. Your goal is
    to generate comprehensive Q/A pairs that cover the full scope of the
     chapter while emphasizing the specified key factual elements.

SOURCE MATERIAL:
Here is the text to analyze:
data_tuple[1]

KEY FACTUAL ELEMENTS (PRIORITY FOCUS):
The key factual elements that MUST be incorporated into answers
    whenever relevant are:
- Date: data_tuple[0][0]
- Location: data_tuple[0][1]
- Person/Entity: data_tuple[0][2]
- Event/Topic: data_tuple[0][3]
```

Table 5: Comparison of recollection strategies in Generative Cued Replay (GCR). Each strategy queries the model's parametric memory to retrieve related episodes before learning a new event.

| Current Event (t, s, e, c) | GCR-Generated (LLM Queries) | GCR-Simple (4 Templates) | GCR-Rich (Detailed Templates) |
|---|---|---|---|
| At Washington Square Park on February 27, 2026, Henry Reed unveiled his groundbreaking designs. Models strutted, showcasing futuristic attire. Henry revealed future collections, leaving the audience in awe. | **1.** What do you remember about Henry Reed's background or previous activities in this narrative, if anything? : **N.** Have any specific models been mentioned earlier in the story? If so, what do you remember about them? | **1.** List everything you remember about February 27, 2026 in the book. **2.** List everything you remember in this book about Washington Square Park. **3.** List everything you remember in this book about Henry Reed. **4.** List everything you remember in this book about Fashion Show. | **1.** Recall all the events that occurred on February 27, 2026. Without describing the events, list all the unique locations where these events took place. : **12.** Think about Henry Reed's experiences. Describe all the key events they've been involved in, focusing on what happened rather than when or where it occurred. |

```
- Key Detail: data_tuple[0][4]

INSTRUCTIONS:
Generate exactly 20 question-answer pairs following these guidelines:

QUESTION REQUIREMENTS:
1. Variety: Include different question types (factual, analytical,
   comparative, causal, definitional)
2. Complexity Range: Mix simple recall questions (30%) with more
   complex analytical questions (70%)
3. Key Element Integration: At least 10 questions should directly
   reference the key factual elements above
4. Comprehensive Coverage: Questions should span the entire chapter,
   not just focus on one section
5. Natural Language: Questions should sound like they come from a human
    learner or teacher

ANSWER REQUIREMENTS:
1. SOURCE-BASED ONLY: Answers must be EXCLUSIVELY based on information
    found in the provided chapter text. DO NOT invent, assume, or add
   any information not explicitly stated in the source material.
2. FACTUAL RECALL: All answers must be direct recalls from the chapter
   - no speculation, inference beyond what's clearly stated, or
   external knowledge.
3. MANDATORY ENTITY INCLUSION: Every answer must incorporate at least
   one of the key factual elements (date, location, person/entity,
   event/topic, key detail) when relevant to the question.
4. VERBATIM ACCURACY: When referencing specific facts, dates, names, or
    details, use the exact information as presented in the source text.
5. NO FABRICATION: If information to answer a question is not available
    in the chapter, do not create that question.
6. Length: Aim for 2-4 sentences per answer, but prioritize accuracy
   over length.

QUESTION TYPE DISTRIBUTION:
- Factual Recall (10 questions): Who, what, when, where questions
- Analytical (5 questions): Why, how, explain, analyze questions
```

```
 - Comparative (2 questions): Compare, contrast, similarities/
     differences
 - Application (2 questions): What would happen if, how would this apply
 - Synthesis (1 question): Summarize, conclude, overall significance

MANDATORY OUTPUT FORMAT:
You must output ONLY the Python list of dictionaries, with no
    additional text, explanations, or formatting. Do not include any
    introductory text like "Here are the question-answer pairs" or any
    closing remarks. Your response should start with [ and end with ].
    Each dictionary must have 'question' and 'correct_answer' keys.

Format exactly like this:
[{"question": "Question 1", "correct_answer": "Answer 1"}, {"question":
     "Question 2", "correct_answer": "Answer 2"}, ...]

CRITICAL CONSTRAINTS:
 - ONLY use information explicitly stated in the provided chapter text
 - DO NOT add external knowledge or make assumptions beyond what's
     written
 - EVERY answer must reference at least one key factual element (date:
     data_tuple[0][0], location: data_tuple[0][1], person/entity:
     data_tuple[0][2], event/topic: data_tuple[0][3], or key detail:
     data_tuple[0][4])
 - If you cannot answer a question based solely on the chapter content,
     do not include that question
 - Answers must be factual recalls, not interpretations or
     extrapolations
 - All 15 Q/A pairs must be created
 - Key factual elements must be prominently featured
 - Full chapter coverage must be achieved
 - Question types must be varied appropriately
 - Answers must be accurate and complete
 - Natural, educational language must be used throughout

Generate the 20 Q/A pairs now in the specified Python list format.
```

Listing 12: Question-answer pair generation prompt template for fine-tuning

This prompt generates questions exemplified in Table 6, which includes both the source chapter corpus and ground-truth questions for comparison.

Table 6: Comparison of question generation strategies for single-event queries (SEQ). The LLM-Generated Asker creates natural, diverse questions while Ground-Truth templates follow fixed patterns.

| Event | LLM-Generated SEQ | Ground-Truth SEQ |
|---|---|---|
| At the Statue of Liberty on June 14, 2025, Brooklyn Ross captivated the audience. Amidst the iconic backdrop, she explained fabric choices with precision. The fashion show attendees hung on her every word. | Who was the central figure captivating the audience at the Statue of Liberty? When and where did Brooklyn Ross give her presentation? What type of event was taking place at the Statue of Liberty? | Provide a comprehensive account of what happened involving Brooklyn Ross and Fashion Show at Statue of Liberty on June 14, 2025. Include all relevant details about the event(s), including what occurred and any other pertinent information. |

## A.7 MERGE

Here we present the prompt template for merging content-based and recollection question-answer pairs. As shown in Listing 13, the merger filters out negative recollection responses, identifies shared entities between question sets, and combines questions that demonstrate meaningful episodic connections while preserving the original number of content-based questions.

```
CONTEXT INFO: context

TASK:
You are tasked with merging two sets of Q&A pairs: content-based
    questions and recollection questions. The output must preserve the
    total number of content-based questions. Follow these specific rules
    :

FILTERING RULES:
1. **Filter out recollection questions** where the answer indicates:
   - "This information is not present"
   - "I haven't encountered this before"
   - "No previous instances found"
   - "Not mentioned in my training"
   - Any similar negative responses indicating lack of prior knowledge

2. **Keep only recollection questions** that demonstrate actual
    connections to previous knowledge

MERGING RULES:
3. **Identify shared entities** between content and recollection
    questions:
   - Same places/locations
   - Same dates/time periods
   - Same entities/characters
   - Same events/actions
   - Same concepts/objects
   - Any other proven connections

4. **For content questions with shared entities in recollection
    questions**:
   - Only merge when there are clear, substantial connections that
    contains potential conflicts of information in the 2 questions
   - Transform them into broader questions that encompass both the
    content information AND the recollection connections
   - Create comprehensive answers that incorporate timeline and context
     from recollection data
   - Create questions that test understanding of both the current
    content and its relationship to previous knowledge
   - Be conservative in identifying shared entities - only merge when
    the connection adds meaningful context, otherwise keep the original
    questions as they are.

5. **For content questions without matching recollection connections**:
   - Keep the original content-based questions exactly as they are
   - Do not modify questions that have no recollection counterparts
   - Preserve original questions when recollection connections are
    minor, superficial, and don't generate any kind of conflict

QUALITY GUIDELINES:
- Merged questions should be clear and well-formed
- Answers should incorporate both current and recalled information when
     merging
- Remove redundancy while preserving important details
- Fix any grammatical errors or awkward phrasing
- Ensure the final output contains at least the same number of
    questions as the content-based dataset
```

```
1566   CRITICAL: You MUST output at least content_count questions (the exact
1567       number of content-based questions), NO LESS!
1568
1569   INPUT DATA:format_qa_dataframe(df1, "Content-based questions")
1570       format_qa_dataframe(df2, "Recollection questions")
1571
1572   REQUIRED JSON FORMAT:
1573   [
1574       {"question": "Your question here", "correct_answer": "Your answer
1575       here"},
1576       {"question": "Your question here", "correct_answer": "Your answer
1577       here"},
1578       ...
1579   ]
1580
1581   IMPORTANT JSON RULES:
1582   - Start with [ and end with ]
1583   - Each object must have exactly "question" and "correct_answer" keys
1584   - Use double quotes for all strings
1585   - Separate objects with commas
1586   - No trailing comma after the last object
1587   - Escape any quotes inside strings with backslashes
1588
1589   OUTPUT ONLY THE JSON ARRAY - NO OTHER TEXT WHATSOEVER.
1590
1591   WARNING: Outputting fewer questions than the content-based dataset is
1592       considered a failure!!
```

Listing 13: Q&A merging prompt template for combining content-based and recollection questions

## A.8 FILTER

Here we present the prompt template used for hallucination filtering of recollection answers. Each recollection response is compared against the source text to remove unsupported information while preserving only content that can be verified from the provided narrative corpus. The filter removes fabricated details and contradictions but does not add missing information when the original answer is incomplete, ensuring that only grounded episodic recall is retained for training.

```
1600   You are an expert fact-checker tasked with removing hallucinations from
1601       an AI model's answer based on provided source text.
1602
1603   **Your Task:**
1604   1. Compare the AI's answer against the given source text
1605   2. Remove any information that is NOT supported by or contradicts the
1606       source text
1607   3. Keep only the parts that are accurate and grounded in the source
1608       material
1609   4. If the question asks about topics not covered in the source text,
1610       return the fallback response: "I have no information to respond to
1611       this question."
1612   5. Return the cleaned answer in the exact format specified below
1613
1614   **Critical Output Requirements:**
1615   - Return ONLY a clean, readable paragraph or short paragraphs
1616   - NO bullet points, NO numbered lists, NO markdown formatting
1617   - NO asterisks (*), NO dashes (-), NO special characters for formatting
1618   - Use plain text with proper sentences and periods
1619   - Keep answers comprehensive but remove redundancy
       - If multiple facts, separate them with periods in flowing sentences
       - If the entire answer becomes invalid, return EXACTLY: "I have no
           information to respond to this question."

       **Content Filtering Rules:**
```

```
- Only retain information directly verifiable from the source text
- Remove invented facts, fictional details, or unsupported claims
- Remove repetitive or redundant information
- Preserve original phrasing when possible for retained content
- Do not add new information beyond what's in the original answer
- Focus on filtering the original answer, not rewriting it completely
- If uncertain about a claim, remove it entirely

**Response Format Example:**
Clean, readable paragraph format with proper sentences and periods. Use
    plain text only.

**Source Text:**
text_up_to

**Question Being Asked:**
question

**AI Answer to Clean:**
answer_llm

**Your Response (cleaned answer only):**
```

Listing 14: Hallucination filtering prompt template for recollection answer validation

### A.9 O-LoRa Implementation Details

Although O-LoRA is designed for task-incremental continual learning, we include it as a baseline because its orthogonality constraints provide a relevant point of comparison for preventing interference between sequential updates. In our adaptation, each event is treated operationally as a task solely for the purpose of applying O-LoRA's update rules, with orthogonal weight updates intended to prevent catastrophic forgetting of previously learned events. However, our setting differs from the original formulation in two key aspects: events do not constitute distinct tasks in the traditional sense, and our per-event training samples are considerably smaller than typical task-incremental datasets.

To ensure a fair comparison, we performed a grid search over O-LoRA's regularization hyperparameters $\lambda_1$ and $\lambda_2$ to account for these differences. We used the implementation from Chen (2024).

## B Results

### B.1 GridSearch

#### B.1.1 10 Events

Table 7: GridSearch Results - 10 Events with Continual-GT Baseline

| Event per Question | Model | Learning Rate | Epochs | Batch Size | Count | Episodic Accuracy |
|---|---|---|---|---|---|---|
| 1 | Llama3-3B | 1e-4 | 5 | 16 | 121 | 2.48 |
| 1 | Llama3-3B | 1e-4 | 20 | 16 | 121 | 42.98 |
| 1 | Llama3-3B | 1e-4 | 30 | 16 | 121 | 29.75 |
| 1 | Llama3-3B | 1e-5 | 5 | 16 | 121 | 0.00 |
| 1 | Llama3-3B | 1e-5 | 20 | 16 | 121 | 9.92 |
| 1 | Llama3-3B | 1e-5 | 10 | 32 | 121 | 0.00 |
| 1 | Llama3-8B | 1e-5 | 20 | 16 | 121 | 34.71 |
| 1 | Llama3-8B | 1e-5 | 40 | 16 | 121 | 42.15 |
| 1 | Llama3-8B | 1e-5 | 60 | 16 | 121 | 42.15 |
| 1 | Llama3-8B | 1e-6 | 80 | 16 | 121 | 40.50 |
| 1 | Llama3-8B | 1e-6 | 100 | 16 | 121 | 26.45 |
| 1 | Llama3-8B | 1e-6 | 120 | 16 | 121 | 33.88 |
| 1 | Llama2-13B | 1e-5 | 20 | 8 | 121 | 28.10 |
| 1 | Llama2-13B | 1e-5 | 40 | 8 | 121 | 38.02 |
| 1 | Llama2-13B | 1e-6 | 20 | 8 | 121 | 1.65 |
| 1 | Llama2-13B | 1e-6 | 40 | 8 | 121 | 1.65 |
| 1 | Llama3-70B | 1e-5 | 10 | 1 | 121 | 38.02 |
| 1 | Llama3-70B | 1e-5 | 20 | 1 | 121 | 33.06 |
| 1 | Llama3-70B | 1e-5 | 40 | 1 | 121 | 28.93 |
| 1 | Llama3-70B | 1e-6 | 10 | 1 | 121 | 0.00 |
| 1 | Llama3-70B | 1e-6 | 20 | 1 | 121 | 23.97 |
| 1 | Llama3-70B | 1e-6 | 40 | 1 | 121 | 33.88 |
| 1 | Llama3-70B | 1e-5 | 20 | 4 | 121 | 42.98 |
| 1 | Llama3-70B | 1e-5 | 40 | 4 | 121 | 42.15 |
| 1 | Llama3-70B | 1e-6 | 20 | 4 | 121 | 27.27 |
| 2 | Llama3-3B | 1e-4 | 5 | 16 | 17 | 0.00 |
| 2 | Llama3-3B | 1e-4 | 20 | 16 | 17 | 23.53 |
| 2 | Llama3-3B | 1e-4 | 30 | 16 | 17 | 5.88 |
| 2 | Llama3-3B | 1e-5 | 5 | 16 | 17 | 0.00 |
| 2 | Llama3-3B | 1e-5 | 20 | 16 | 17 | 5.88 |
| 2 | Llama3-3B | 1e-5 | 10 | 32 | 17 | 0.00 |
| 2 | Llama3-8B | 1e-5 | 20 | 16 | 17 | 17.65 |
| 2 | Llama3-8B | 1e-5 | 40 | 16 | 17 | 29.41 |
| 2 | Llama3-8B | 1e-5 | 60 | 16 | 17 | 29.41 |
| 2 | Llama3-8B | 1e-6 | 80 | 16 | 17 | 29.41 |
| 2 | Llama3-8B | 1e-6 | 100 | 16 | 17 | 41.18 |
| 2 | Llama3-8B | 1e-6 | 120 | 16 | 17 | 35.29 |
| 2 | Llama2-13B | 1e-5 | 20 | 8 | 17 | 11.76 |
| 2 | Llama2-13B | 1e-5 | 40 | 8 | 17 | 5.88 |
| 2 | Llama2-13B | 1e-6 | 20 | 8 | 17 | 0.00 |
| 2 | Llama2-13B | 1e-6 | 40 | 8 | 17 | 0.00 |
| 2 | Llama3-70B | 1e-5 | 10 | 1 | 17 | 23.53 |
| 2 | Llama3-70B | 1e-5 | 20 | 1 | 17 | 29.41 |
| 2 | Llama3-70B | 1e-5 | 40 | 1 | 17 | 5.88 |
| 2 | Llama3-70B | 1e-6 | 10 | 1 | 17 | 0.00 |
| 2 | Llama3-70B | 1e-6 | 20 | 1 | 17 | 5.88 |
| 2 | Llama3-70B | 1e-6 | 40 | 1 | 17 | 23.53 |
| 2 | Llama3-70B | 1e-5 | 20 | 4 | 17 | 29.41 |
| 2 | Llama3-70B | 1e-5 | 40 | 4 | 17 | 29.41 |
| 2 | Llama3-70B | 1e-6 | 20 | 4 | 17 | 11.76 |
| 3-5 | Llama3-3B | 1e-4 | 5 | 16 | 21 | 4.76 |
| 3-5 | Llama3-3B | 1e-4 | 20 | 16 | 21 | 19.05 |
| 3-5 | Llama3-3B | 1e-4 | 30 | 16 | 21 | 0.00 |
| 3-5 | Llama3-3B | 1e-5 | 5 | 16 | 21 | 0.00 |
| 3-5 | Llama3-3B | 1e-5 | 20 | 16 | 21 | 4.76 |
| 3-5 | Llama3-3B | 1e-5 | 10 | 32 | 21 | 0.00 |
| 3-5 | Llama3-8B | 1e-5 | 20 | 16 | 21 | 4.76 |
| 3-5 | Llama3-8B | 1e-5 | 40 | 16 | 21 | 14.29 |
| 3-5 | Llama3-8B | 1e-5 | 60 | 16 | 21 | 9.52 |
| 3-5 | Llama3-8B | 1e-6 | 80 | 16 | 21 | 14.29 |
| 3-5 | Llama3-8B | 1e-6 | 100 | 16 | 21 | 4.76 |
| 3-5 | Llama3-8B | 1e-6 | 120 | 16 | 21 | 9.52 |
| 3-5 | Llama2-13B | 1e-5 | 20 | 8 | 21 | 4.76 |
| 3-5 | Llama2-13B | 1e-5 | 40 | 8 | 21 | 9.52 |
| 3-5 | Llama2-13B | 1e-6 | 20 | 8 | 21 | 0.00 |
| 3-5 | Llama2-13B | 1e-6 | 40 | 8 | 21 | 0.00 |
| 3-5 | Llama3-70B | 1e-5 | 10 | 1 | 21 | 9.52 |
| 3-5 | Llama3-70B | 1e-5 | 20 | 1 | 21 | 14.29 |
| 3-5 | Llama3-70B | 1e-5 | 40 | 1 | 21 | 0.00 |
| 3-5 | Llama3-70B | 1e-6 | 10 | 1 | 21 | 0.00 |
| 3-5 | Llama3-70B | 1e-6 | 20 | 1 | 21 | 4.76 |
| 3-5 | Llama3-70B | 1e-6 | 40 | 1 | 21 | 4.76 |
| 3-5 | Llama3-70B | 1e-5 | 20 | 4 | 21 | 19.05 |
| 3-5 | Llama3-70B | 1e-5 | 40 | 4 | 21 | 9.52 |
| 3-5 | Llama3-70B | 1e-6 | 20 | 4 | 21 | 0.00 |

## B.1.2 30 EVENTS

Table 8: GridSearch Results - 30 Events with Continual-GT Baseline

| Event per Question | Model | Learning Rate | Epochs | Batch Size | Count | Episodic Accuracy |
|---|---|---|---|---|---|---|
| 1 | Llama3-3B | 1e-4 | 20 | 16 | 126 | 8.73 |
| 1 | Llama3-3B | 1e-4 | 40 | 16 | 126 | 11.90 |
| 1 | Llama3-3B | 1e-5 | 20 | 16 | 126 | 8.73 |
| 1 | Llama3-3B | 1e-5 | 40 | 16 | 126 | 10.32 |
| 1 | Llama3-8B | 1e-5 | 20 | 16 | 126 | 18.25 |
| 1 | Llama3-8B | 1e-5 | 40 | 16 | 126 | 17.46 |
| 1 | Llama3-8B | 1e-6 | 20 | 16 | 126 | 0.00 |
| 1 | Llama3-8B | 1e-6 | 40 | 16 | 126 | 10.32 |
| 1 | Llama2-13B | 1e-5 | 20 | 8 | 126 | 9.52 |
| 1 | Llama2-13B | 1e-5 | 40 | 8 | 126 | 10.32 |
| 1 | Llama2-13B | 1e-6 | 20 | 8 | 126 | 0.79 |
| 1 | Llama2-13B | 1e-6 | 40 | 8 | 126 | 1.59 |
| 1 | Llama3-70B | 1e-5 | 40 | 1 | 126 | 14.29 |
| 1 | Llama3-70B | 1e-5 | 20 | 4 | 126 | 11.90 |
| 1 | Llama3-70B | 1e-5 | 40 | 4 | 126 | 12.70 |
| 1 | Llama3-70B | 1e-6 | 20 | 4 | 126 | 10.32 |
| 1 | Llama3-70B | 1e-6 | 40 | 4 | 126 | 11.90 |
| 2 | Llama3-3B | 1e-4 | 20 | 16 | 42 | 0.00 |
| 2 | Llama3-3B | 1e-4 | 40 | 16 | 42 | 4.76 |
| 2 | Llama3-3B | 1e-5 | 20 | 16 | 42 | 2.38 |
| 2 | Llama3-3B | 1e-5 | 40 | 16 | 42 | 0.00 |
| 2 | Llama3-8B | 1e-5 | 20 | 16 | 42 | 9.52 |
| 2 | Llama3-8B | 1e-5 | 40 | 16 | 42 | 11.90 |
| 2 | Llama3-8B | 1e-6 | 20 | 16 | 42 | 0.00 |
| 2 | Llama3-8B | 1e-6 | 40 | 16 | 42 | 2.38 |
| 2 | Llama2-13B | 1e-5 | 20 | 8 | 42 | 2.38 |
| 2 | Llama2-13B | 1e-5 | 40 | 8 | 42 | 0.00 |
| 2 | Llama2-13B | 1e-6 | 20 | 8 | 42 | 0.00 |
| 2 | Llama2-13B | 1e-6 | 40 | 8 | 42 | 0.00 |
| 2 | Llama3-70B | 1e-5 | 40 | 1 | 42 | 11.90 |
| 2 | Llama3-70B | 1e-5 | 20 | 4 | 42 | 2.38 |
| 2 | Llama3-70B | 1e-5 | 40 | 4 | 42 | 2.38 |
| 2 | Llama3-70B | 1e-6 | 20 | 4 | 42 | 0.00 |
| 2 | Llama3-70B | 1e-6 | 40 | 4 | 42 | 2.38 |
| 3-5 | Llama3-3B | 1e-4 | 20 | 16 | 53 | 0.00 |
| 3-5 | Llama3-3B | 1e-4 | 40 | 16 | 53 | 0.00 |
| 3-5 | Llama3-3B | 1e-5 | 20 | 16 | 53 | 0.00 |
| 3-5 | Llama3-3B | 1e-5 | 40 | 16 | 53 | 1.89 |
| 3-5 | Llama3-8B | 1e-5 | 20 | 16 | 53 | 1.89 |
| 3-5 | Llama3-8B | 1e-5 | 40 | 16 | 53 | 3.77 |
| 3-5 | Llama3-8B | 1e-6 | 20 | 16 | 53 | 0.00 |
| 3-5 | Llama3-8B | 1e-6 | 40 | 16 | 53 | 3.77 |
| 3-5 | Llama2-13B | 1e-5 | 20 | 8 | 53 | 3.77 |
| 3-5 | Llama2-13B | 1e-5 | 40 | 8 | 53 | 0.00 |
| 3-5 | Llama2-13B | 1e-6 | 20 | 8 | 53 | 0.00 |
| 3-5 | Llama2-13B | 1e-6 | 40 | 8 | 53 | 0.00 |
| 3-5 | Llama3-70B | 1e-5 | 40 | 1 | 53 | 9.43 |
| 3-5 | Llama3-70B | 1e-5 | 20 | 4 | 53 | 1.89 |
| 3-5 | Llama3-70B | 1e-5 | 40 | 4 | 53 | 0.00 |
| 3-5 | Llama3-70B | 1e-6 | 20 | 4 | 53 | 0.00 |
| 3-5 | Llama3-70B | 1e-6 | 40 | 4 | 53 | 0.00 |
| 6+ | Llama3-3B | 1e-4 | 20 | 16 | 18 | 0.00 |
| 6+ | Llama3-3B | 1e-4 | 40 | 16 | 18 | 0.00 |
| 6+ | Llama3-3B | 1e-5 | 20 | 16 | 18 | 0.00 |
| 6+ | Llama3-3B | 1e-5 | 40 | 16 | 18 | 0.00 |
| 6+ | Llama3-8B | 1e-5 | 20 | 16 | 18 | 0.00 |
| 6+ | Llama3-8B | 1e-5 | 40 | 16 | 18 | 5.56 |
| 6+ | Llama3-8B | 1e-6 | 20 | 16 | 18 | 0.00 |
| 6+ | Llama3-8B | 1e-6 | 40 | 16 | 18 | 0.00 |
| 6+ | Llama2-13B | 1e-5 | 20 | 8 | 18 | 0.00 |
| 6+ | Llama2-13B | 1e-5 | 40 | 8 | 18 | 0.00 |
| 6+ | Llama2-13B | 1e-6 | 20 | 8 | 18 | 0.00 |
| 6+ | Llama2-13B | 1e-6 | 40 | 8 | 18 | 0.00 |
| 6+ | Llama3-70B | 1e-5 | 40 | 1 | 18 | 0.00 |
| 6+ | Llama3-70B | 1e-5 | 20 | 4 | 18 | 0.00 |
| 6+ | Llama3-70B | 1e-5 | 40 | 4 | 18 | 0.00 |
| 6+ | Llama3-70B | 1e-6 | 20 | 4 | 18 | 0.00 |
| 6+ | Llama3-70B | 1e-6 | 40 | 4 | 18 | 0.00 |

## B.1.3 100 EVENTS

Table 9: GridSearch Results - 100 Events with Continual-GT Baseline

| Event per Question | Model | Learning Rate | Epochs | Batch Size | Count | Episodic Accuracy |
|---|---|---|---|---|---|---|
| 1 | Llama3-3B | 1e-4 | 20 | 16 | 135 | 11.85 |
| 1 | Llama3-3B | 1e-4 | 40 | 16 | 135 | 8.15 |
| 1 | Llama3-3B | 1e-5 | 20 | 16 | 135 | 4.44 |
| 1 | Llama3-3B | 1e-5 | 40 | 16 | 135 | 13.33 |
| 1 | Llama3-8B | 1e-5 | 20 | 16 | 135 | 10.37 |
| 1 | Llama3-8B | 1e-5 | 40 | 16 | 135 | 22.96 |
| 1 | Llama3-8B | 1e-6 | 20 | 16 | 135 | 0.00 |
| 1 | Llama3-8B | 1e-6 | 40 | 16 | 135 | 7.41 |
| 1 | Llama2-13B | 1e-5 | 20 | 8 | 135 | 4.44 |
| 1 | Llama2-13B | 1e-5 | 40 | 8 | 135 | 4.44 |
| 1 | Llama2-13B | 1e-6 | 20 | 8 | 135 | 2.96 |
| 1 | Llama2-13B | 1e-6 | 40 | 8 | 135 | 2.22 |
| 1 | Llama3-70B | 1e-5 | 20 | 1 | 135 | 22.22 |
| 1 | Llama3-70B | 1e-5 | 20 | 4 | 135 | 7.41 |
| 1 | Llama3-70B | 1e-5 | 40 | 4 | 135 | 11.11 |
| 1 | Llama3-70B | 1e-6 | 20 | 4 | 135 | 13.33 |
| 1 | Llama3-70B | 1e-6 | 40 | 4 | 135 | 13.33 |
| 2 | Llama3-3B | 1e-4 | 20 | 16 | 78 | 7.69 |
| 2 | Llama3-3B | 1e-4 | 40 | 16 | 78 | 0.00 |
| 2 | Llama3-3B | 1e-5 | 20 | 16 | 78 | 1.28 |
| 2 | Llama3-3B | 1e-5 | 40 | 16 | 78 | 7.69 |
| 2 | Llama3-8B | 1e-5 | 20 | 16 | 78 | 6.41 |
| 2 | Llama3-8B | 1e-5 | 40 | 16 | 78 | 14.10 |
| 2 | Llama3-8B | 1e-6 | 20 | 16 | 78 | 2.56 |
| 2 | Llama3-8B | 1e-6 | 40 | 16 | 78 | 3.85 |
| 2 | Llama2-13B | 1e-5 | 20 | 8 | 78 | 1.28 |
| 2 | Llama2-13B | 1e-5 | 40 | 8 | 78 | 2.56 |
| 2 | Llama2-13B | 1e-6 | 20 | 8 | 78 | 1.28 |
| 2 | Llama2-13B | 1e-6 | 40 | 8 | 78 | 1.28 |
| 2 | Llama3-70B | 1e-5 | 20 | 1 | 78 | 19.23 |
| 2 | Llama3-70B | 1e-5 | 20 | 4 | 78 | 1.28 |
| 2 | Llama3-70B | 1e-5 | 40 | 4 | 78 | 5.13 |
| 2 | Llama3-70B | 1e-6 | 20 | 4 | 78 | 0.00 |
| 2 | Llama3-70B | 1e-6 | 40 | 4 | 78 | 7.69 |
| 3-5 | Llama3-3B | 1e-4 | 20 | 16 | 81 | 1.23 |
| 3-5 | Llama3-3B | 1e-4 | 40 | 16 | 81 | 0.00 |
| 3-5 | Llama3-3B | 1e-5 | 20 | 16 | 81 | 0.00 |
| 3-5 | Llama3-3B | 1e-5 | 40 | 16 | 81 | 0.00 |
| 3-5 | Llama3-8B | 1e-5 | 20 | 16 | 81 | 2.47 |
| 3-5 | Llama3-8B | 1e-5 | 40 | 16 | 81 | 0.00 |
| 3-5 | Llama3-8B | 1e-6 | 20 | 16 | 81 | 0.00 |
| 3-5 | Llama3-8B | 1e-6 | 40 | 16 | 81 | 1.23 |
| 3-5 | Llama2-13B | 1e-5 | 20 | 8 | 81 | 0.00 |
| 3-5 | Llama2-13B | 1e-5 | 40 | 8 | 81 | 0.00 |
| 3-5 | Llama2-13B | 1e-6 | 20 | 8 | 81 | 0.00 |
| 3-5 | Llama2-13B | 1e-6 | 40 | 8 | 81 | 0.00 |
| 3-5 | Llama3-70B | 1e-5 | 20 | 1 | 81 | 14.81 |
| 3-5 | Llama3-70B | 1e-5 | 20 | 4 | 81 | 0.00 |
| 3-5 | Llama3-70B | 1e-5 | 40 | 4 | 81 | 0.00 |
| 3-5 | Llama3-70B | 1e-6 | 20 | 4 | 81 | 3.70 |
| 3-5 | Llama3-70B | 1e-6 | 40 | 4 | 81 | 4.94 |
| 6+ | Llama3-3B | 1e-4 | 20 | 16 | 63 | 0.00 |
| 6+ | Llama3-3B | 1e-4 | 40 | 16 | 63 | 0.00 |
| 6+ | Llama3-3B | 1e-5 | 20 | 16 | 63 | 0.00 |
| 6+ | Llama3-3B | 1e-5 | 40 | 16 | 63 | 0.00 |
| 6+ | Llama3-8B | 1e-5 | 20 | 16 | 63 | 1.59 |
| 6+ | Llama3-8B | 1e-5 | 40 | 16 | 63 | 0.00 |
| 6+ | Llama3-8B | 1e-6 | 20 | 16 | 63 | 0.00 |
| 6+ | Llama3-8B | 1e-6 | 40 | 16 | 63 | 0.00 |
| 6+ | Llama2-13B | 1e-5 | 20 | 8 | 63 | 0.00 |
| 6+ | Llama2-13B | 1e-5 | 40 | 8 | 63 | 0.00 |
| 6+ | Llama2-13B | 1e-6 | 20 | 8 | 63 | 0.00 |
| 6+ | Llama2-13B | 1e-6 | 40 | 8 | 63 | 0.00 |
| 6+ | Llama3-70B | 1e-5 | 20 | 1 | 63 | 1.59 |
| 6+ | Llama3-70B | 1e-5 | 20 | 4 | 63 | 0.00 |
| 6+ | Llama3-70B | 1e-5 | 40 | 4 | 63 | 0.00 |
| 6+ | Llama3-70B | 1e-6 | 20 | 4 | 63 | 4.76 |
| 6+ | Llama3-70B | 1e-6 | 40 | 4 | 63 | 1.59 |

# C  SCALING EFFECTS

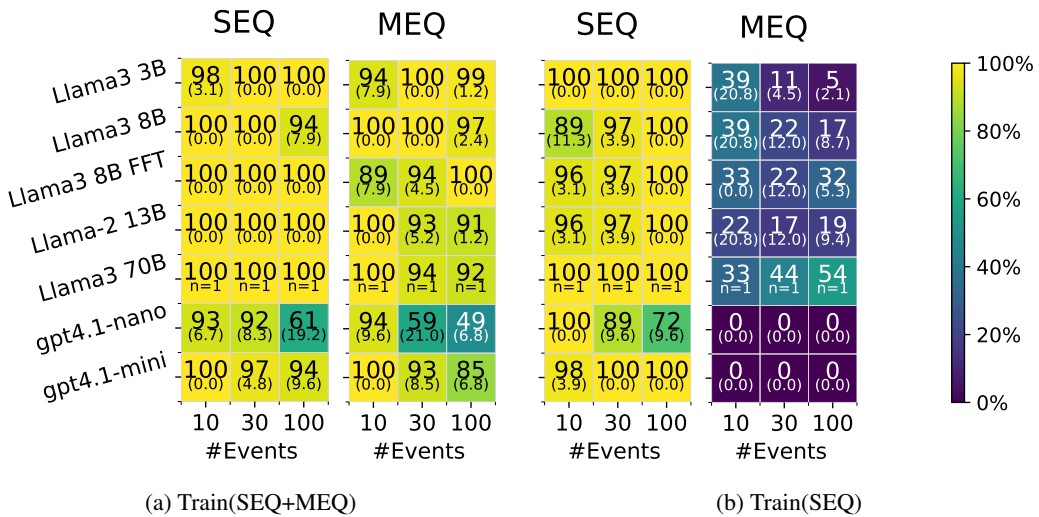

(a) Train(SEQ+MEQ)  (b) Train(SEQ)

Figure 6: Average (standard deviation) performance using lenient recall metric for chronological ordering over 3 books. Standard deviations are calculated across books.

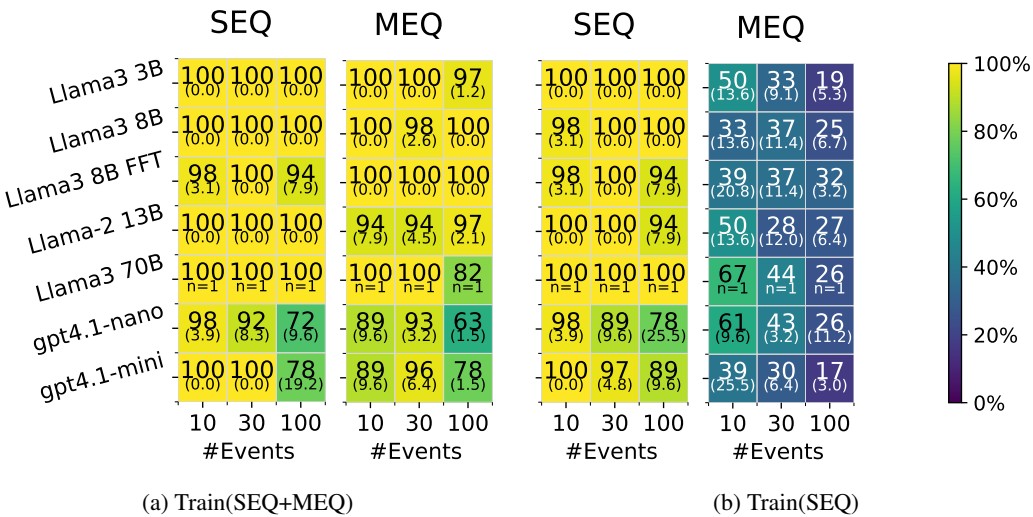

(a) Train(SEQ+MEQ)  (b) Train(SEQ)

Figure 7: Average (standard deviation) performance using lenient recall metric for latest ordering over 3 books. Standard deviations are calculated across books.

## C.1    IS BINDING LEARNABLE?

### C.1.1    EXPERIMENT 1 - GROKKING

To address concerns about whether longer training could resolve the binding challenge, we conducted extended training experiments with Llama-3-8B and a 100 events narrative, pushing the model up to 3000 epochs. This experiment was inspired by recent observations that extended training can lead to delayed generalization phenomena such as grokking (Power et al., 2022).

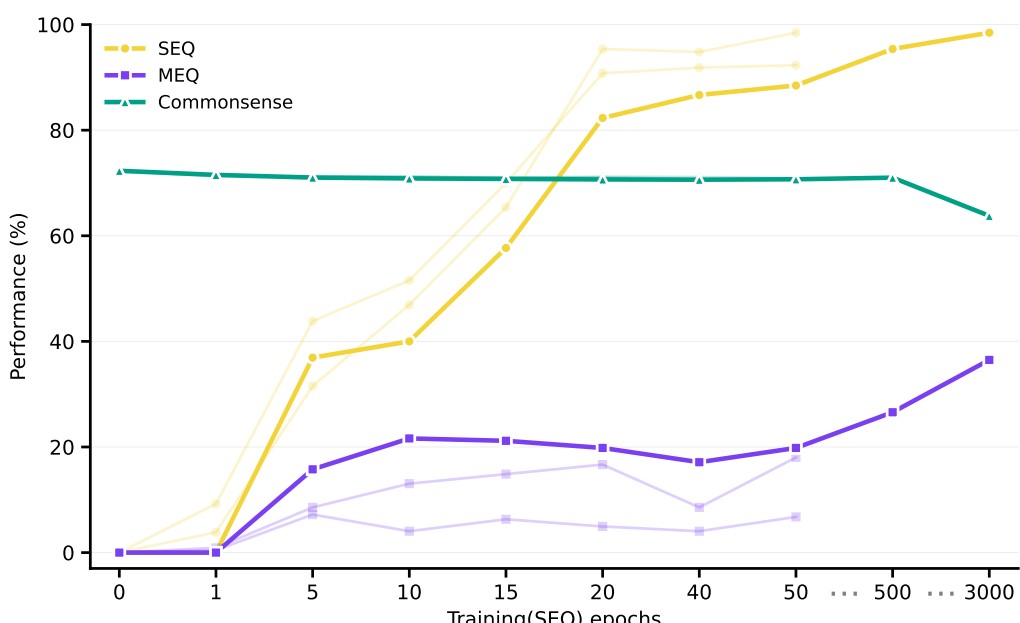

Figure 8: Performance of Llama-3-8B across training epochs on three narrative universes (default, scifi, and news). SEQ accuracy reaches near-perfect performance while MEQ remains poor even after 3000 epochs. The slight MEQ improvement at higher epochs comes at the cost of degraded commonsense performance, indicating the model is overfitting to training examples rather than learning to bind related episodes.

While the model reaches nearly 100% accuracy on SEQ questions (demonstrating successful memorization of individual facts), MEQ performance remains poor across all three universes, even after 3000 epochs of training. This suggests that the binding problem is not simply a matter of insufficient training time or exposure to the data.

We also tracked performance on CommonsenseQA, a general knowledge benchmark with 9K questions, as a sanity check to ensure the model's broader cognitive abilities were not degrading during extended training. Performance remained stable until 500 epochs, but dropped from 70% to 63% at 3000 epochs. This decline, coinciding with marginal MEQ improvements (from ~20% to 37%), indicates the model is overfitting to training examples rather than learning genuine binding. The degradation in general knowledge confirms that extended training beyond 3000 epochs produces memorization artifacts, not improved cross-episode reasoning.

These results reinforce our core claim: current training paradigms struggle to create the cross-episode connections needed for binding, regardless of training duration. The tradeoff between slight MEQ gains and commonsense degradation demonstrates that brute-force training cannot solve the binding problem.

### C.1.2    EXPERIMENT 2 - TRANSFER LEARNING

To investigate whether binding capabilities can transfer across narratives, we conducted cross-narrative experiments with Llama-3-8B using two 100 event narratives. We trained on SEQ of

narrative 1 (we will use this notation: SEQ1), and SEQ + MEQ of narrative 2. Then evaluated on both SEQ1 and MEQ1. This tests whether learning multi-event reasoning patterns from Narrative 2 enables binding in Narrative 1 where only isolated facts were seen during training.

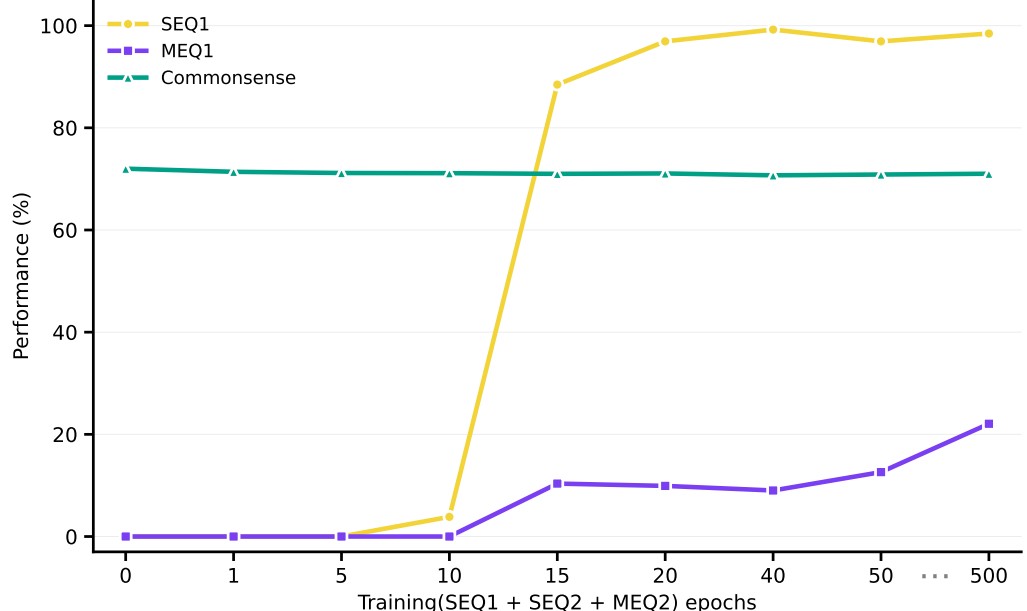

Figure 9: Performance of Llama-3-8B across training epochs of cross-narrative training. This experiment shows that binding do not transfer across narratives.

We tested whether binding patterns could transfer across entities (training on MEQ for 1 narrative while testing on another), but found no such transfer occurs. This validates that binding is not a learnable skill but requires explicit multi-event supervision.

## C.2   GCR ABLATIONS

Table 10: GCR Pipelines Multi Retrieval Task

| #Ev | MSize | Event per question | Continual NoReplay | GCR Simple raw | GCR Simple filtered | GCR Rich raw | GCR Rich filtered | GCR Generated raw | GCR Generated filtered | GT-Baseline |
|---|---|---|---|---|---|---|---|---|---|---|
| 10 | 3B | 1 | 29.75 | 47.11 | 50.41 | 40.50 | 48.76 | 36.36 | 56.20 | 42.98 |
| 10 | 8B | 1 | 28.10 | 43.80 | 29.75 | 44.63 | 41.32 | 52.89 | 40.50 | 42.15 |
| 10 | 3B | 2 | 5.88 | 11.76 | 17.65 | 5.88 | 23.53 | 11.76 | 47.06 | 23.53 |
| 10 | 8B | 2 | 23.53 | 11.76 | 11.76 | 23.53 | 41.18 | 29.41 | 11.76 | 29.41 |
| 10 | 3B | 3-5 | 0.00 | 9.52 | 4.76 | 4.76 | 9.52 | 9.52 | 23.81 | 19.05 |
| 10 | 8B | 3-5 | 0.00 | 4.76 | 0.00 | 38.10 | 19.05 | 14.29 | 4.76 | 14.29 |
| 30 | 3B | 1 | 10.32 | 19.05 | 18.25 | 21.43 | 15.08 | 19.84 | 21.43 | 11.90 |
| 30 | 8B | 1 | 11.11 | 30.16 | 41.27 | 28.57 | 32.54 | 24.60 | 50.00 | 17.46 |
| 30 | 3B | 2 | 0.00 | 7.14 | 11.90 | 9.52 | 9.52 | 4.76 | 9.52 | 4.76 |
| 30 | 8B | 2 | 9.52 | 16.67 | 28.57 | 7.14 | 7.14 | 9.52 | 30.95 | 11.90 |
| 30 | 3B | 3-5 | 0.00 | 1.89 | 1.89 | 1.89 | 0.00 | 0.00 | 1.89 | 0.00 |
| 30 | 8B | 3-5 | 0.00 | 1.89 | 16.98 | 3.77 | 3.77 | 7.55 | 18.87 | 3.77 |
| 30 | 3B | 6+ | 0.00 | 0.00 | 0.00 | 0.00 | 0.00 | 0.00 | 5.56 | 0.00 |
| 30 | 8B | 6+ | 0.00 | 0.00 | 5.56 | 0.00 | 0.00 | 0.00 | 5.56 | 5.56 |
| 100 | 8B | 1 | 9.63 | 25.93 | 25.19 | 20.74 | 35.56 | 22.22 | 39.26 | 22.96 |
| 100 | 8B | 2 | 2.56 | 11.54 | 11.54 | 6.41 | 14.10 | 8.97 | 15.38 | 14.10 |
| 100 | 8B | 3-5 | 0.00 | 1.23 | 2.47 | 1.23 | 2.47 | 2.47 | 11.11 | 0.00 |
| 100 | 8B | 6+ | 0.00 | 0.00 | 0.00 | 0.00 | 1.59 | 0.00 | 6.35 | 0.00 |

Table 11: GCR Pipelines Chronological Task

| #Ev | MSize | Event per question | Continual NoReplay | GCR Simple raw | GCR Simple filtered | GCR Rich raw | GCR Rich filtered | GCR Generated raw | GCR Generated filtered | GT-Baseline |
|---|---|---|---|---|---|---|---|---|---|---|
| 10 | 3B | 1 | 13.33 | 60.00 | 60.00 | 20.00 | 46.67 | 60.00 | 46.67 | 40.00 |
| 10 | 8B | 1 | 20.00 | 46.67 | 20.00 | 66.67 | 26.67 | 60.00 | 40.00 | 46.67 |
| 10 | 3B | 2 | 0.00 | 33.33 | 33.33 | 0.00 | 33.33 | 0.00 | 100.00 | 33.33 |
| 10 | 8B | 2 | 0.00 | 33.33 | 0.00 | 66.67 | 66.67 | 33.33 | 33.33 | 0.00 |
| 10 | 3B | 3-5 | 0.00 | 33.33 | 0.00 | 0.00 | 0.00 | 0.00 | 33.33 | 0.00 |
| 10 | 8B | 3-5 | 0.00 | 0.00 | 33.33 | 33.33 | 66.67 | 0.00 | 33.33 | 0.00 |
| 30 | 3B | 1 | 16.67 | 25.00 | 50.00 | 33.33 | 41.67 | 33.33 | 33.33 | 33.33 |
| 30 | 8B | 1 | 0.00 | 33.33 | 50.00 | 25.00 | 33.33 | 41.67 | 41.67 | 33.33 |
| 30 | 3B | 3-5 | 0.00 | 0.00 | 0.00 | 6.67 | 0.00 | 0.00 | 0.00 | 6.67 |
| 30 | 8B | 3-5 | 0.00 | 6.67 | 26.67 | 0.00 | 6.67 | 13.33 | 20.00 | 13.33 |
| 30 | 3B | 6+ | 0.00 | 0.00 | 0.00 | 0.00 | 0.00 | 0.00 | 0.00 | 0.00 |
| 30 | 8B | 6+ | 0.00 | 0.00 | 0.00 | 0.00 | 0.00 | 0.00 | 0.00 | 33.33 |
| 100 | 8B | 1 | 16.67 | 33.33 | 66.67 | 50.00 | 33.33 | 33.33 | 33.33 | 33.33 |
| 100 | 8B | 2 | 0.00 | 6.67 | 6.67 | 0.00 | 0.00 | 6.67 | 13.33 | 20.00 |
| 100 | 8B | 3-5 | 0.00 | 11.11 | 11.11 | 11.11 | 0.00 | 0.00 | 11.11 | 11.11 |
| 100 | 8B | 6+ | 0.00 | 0.00 | 6.67 | 0.00 | 0.00 | 0.00 | 0.00 | 0.00 |

Table 12: GCR Pipelines Latest Task

| #Ev | MSize | Event per question | Continual NoReplay | GCR Simple raw | GCR Simple filtered | GCR Rich raw | GCR Rich filtered | GCR Generated raw | GCR Generated filtered | GT-Baseline |
|---|---|---|---|---|---|---|---|---|---|---|
| 10 | 3B | 1 | 26.67 | 20.00 | 20.00 | 26.67 | 33.33 | 20.00 | 40.00 | 13.33 |
| 10 | 8B | 1 | 20.00 | 46.67 | 26.67 | 40.00 | 33.33 | 26.67 | 33.33 | 40.00 |
| 10 | 3B | 2 | 0.00 | 0.00 | 0.00 | 0.00 | 0.00 | 0.00 | 0.00 | 0.00 |
| 10 | 8B | 2 | 0.00 | 33.33 | 0.00 | 33.33 | 0.00 | 0.00 | 33.33 | 0.00 |
| 10 | 3B | 3-5 | 0.00 | 0.00 | 0.00 | 0.00 | 0.00 | 0.00 | 0.00 | 0.00 |
| 10 | 8B | 3-5 | 0.00 | 0.00 | 0.00 | 0.00 | 0.00 | 0.00 | 0.00 | 0.00 |
| 30 | 3B | 1 | 16.67 | 16.67 | 16.67 | 0.00 | 25.00 | 8.33 | 8.33 | 33.33 |
| 30 | 8B | 1 | 0.00 | 33.33 | 33.33 | 8.33 | 16.67 | 16.67 | 33.33 | 33.33 |
| 30 | 3B | 3-5 | 13.33 | 6.67 | 33.33 | 6.67 | 26.67 | 13.33 | 26.67 | 26.67 |
| 30 | 8B | 3-5 | 13.33 | 33.33 | 40.00 | 13.33 | 40.00 | 26.67 | 60.00 | 26.67 |
| 30 | 3B | 6+ | 0.00 | 0.00 | 0.00 | 0.00 | 0.00 | 0.00 | 0.00 | 0.00 |
| 30 | 8B | 6+ | 0.00 | 0.00 | 0.00 | 0.00 | 0.00 | 0.00 | 0.00 | 0.00 |
| 100 | 8B | 1 | 16.67 | 16.67 | 33.33 | 16.67 | 33.33 | 0.00 | 33.33 | 33.33 |
| 100 | 8B | 2 | 6.67 | 0.00 | 26.67 | 0.00 | 26.67 | 20.00 | 6.67 | 0.00 |
| 100 | 8B | 3-5 | 11.11 | 0.00 | 0.00 | 0.00 | 33.33 | 22.22 | 0.00 | 0.00 |
| 100 | 8B | 6+ | 6.67 | 13.33 | 20.00 | 6.67 | 20.00 | 6.67 | 6.67 | 6.67 |

### C.2.1 CD PLOT

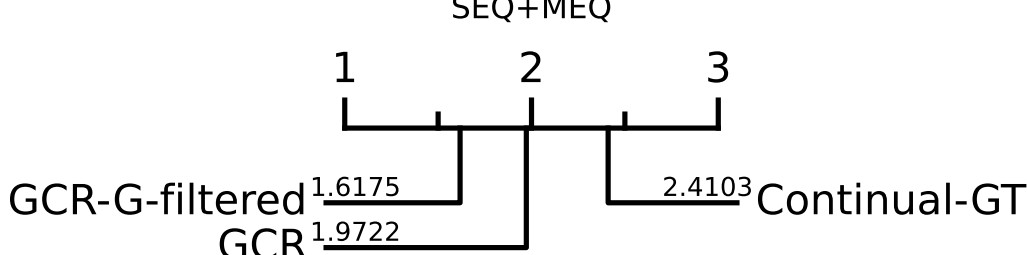

Figure 10: Critical distance analysis for Llama3 8B model performance on a 30-event narrative with 126 SEQ and 113 MEQ questions, comparing Continual-GT pipeline, GCR-G, and GCR-G-filtered approaches.

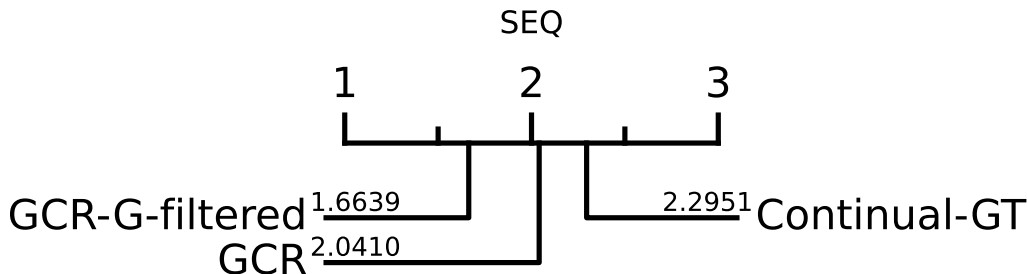

Figure 11: Critical distance analysis for Llama3 8B model performance on a 30-event narrative with 126 SEQ questions, comparing Continual-GT pipeline, GCR-G, and GCR-G-filtered approaches.

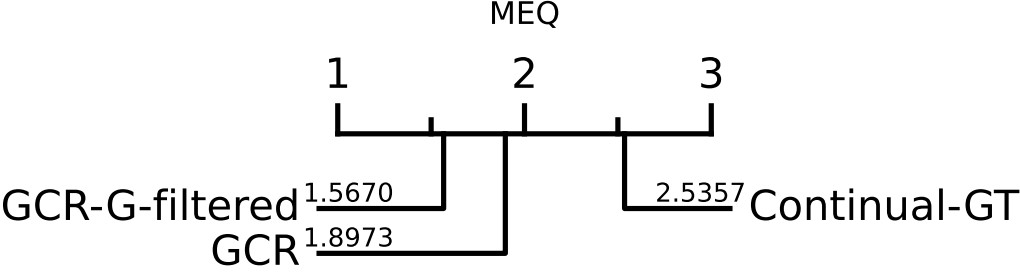

Figure 12: Critical distance analysis for Llama3 8B model performance on a 30-event narrative with 113 MEQ questions, comparing Continual-GT pipeline, GCR-G, and GCR-G-filtered approaches.

## D RELATED WORK EXTENDED

### TAXONOMY DEFINITIONS

Table 13 organizes prior work along five dimensions relevant to our study. We define each as follows:

- **Knowledge Binding**: Whether the work addresses the challenge of linking separately learned facts that share common traits or elements (e.g., retrieving all episodes involving an entity after learning each independently, inferring that A is B after learning that B is A etc.). This is the core problem we touch.

- **Catastrophic Forgetting**: Whether the work studies or mitigates the loss of previously learned information when acquiring new knowledge (a distinct failure mode from binding, as models may retain individual facts yet fail to connect them).
- **Scale Effect**: Whether the work systematically investigates how model size (parameter count) or data scale affects the phenomenon under study.
- **Rehearsal**: Whether the method involves replaying or regenerating past examples during training, either from stored buffers or through generative synthesis.
- **Continual Learning**: Whether the work operates in a sequential learning setting where data arrives over time, as opposed to static (batch) training with all data available upfront.

A checkmark indicates the work's primary contribution addresses that dimension; absence indicates the dimension is not a central focus. Our contribution is the first to jointly address knowledge binding and catastrophic forgetting in a continual learning setting using rehearsal-based methods.

### D.1 CONNECTING ENTITIES ACROSS EPISODES

How language models integrate related but separately learned information remains an open question. Lampinen et al. (2025a) investigated how different learning paradigms affect generalization in syllogistic reasoning tasks, comparing in-context learning and fine-tuning on identical datasets. Their results show that a hybrid approach achieves the best performance, suggesting that the learning mechanism itself plays a crucial role in generalization capabilities.

Other work has explored different facets of the same fundamental problem. Treutlein et al. (2024) demonstrated that models finetuned with LoRa can infer hidden patterns from disparate training examples, successfully identifying latent structures like simple functions or city identities from implicit cues alone, while other work examined how fine-tuning or pretraining affects factual consistency Zucchet et al. (2025). Russin et al. (2025) found compositional generalization emerges only after meta-learning across thousands of tasks. In contrast, Berglund et al. (2023) revealed a systematic failure they termed the "Reversal Curse," where models trained on facts like "A is B" fail to answer "B is A" queries despite perfect forward performance. This limitation persisted across model scales and families, even in GPT-4. Gekhman et al. (2024) found fine-tuning on new knowledge increases hallucinations. Allen-Zhu & Li (2024) demonstrated that LLMs fail at knowledge manipulation tasks like classification and comparison unless explicitly trained with Chain-of-Thought reasoning at both training and inference time.

***Collectively, these works highlight limitations in relational generalization and knowledge transformation, but they do not study how models connect multiple related episodes learned at different times in knowledge representation both in the pre trained model and in the finetuned ones. Here, we systematically study how episodic knowledge binding scales with model size and number of events, verify this phenomenon empirically in both static and continual learning settings, and propose a practical solution for the more challenging and realistic continual learning scenario.***

### D.2 ADDRESSING CATASTROPHIC FORGETTING

The catastrophic forgetting problem has been studied since the early days of neural networks, with foundational work on Elastic Weight Consolidation (Kirkpatrick et al., 2017), experience replay (Rebuffi et al., 2017b), and Progressive Neural Networks (Rusu et al., 2016). Architectural approaches like PackNet (Mallya & Lazebnik, 2018) and Hard Attention to the Task (HAT) (Serra et al., 2018) used weight and attention masking to isolate task-specific parameters. These methods were designed for sequential task learning in computer vision and reinforcement learning, where the goal was preserving performance on previously learned tasks ( already discussed in Section 6 how traditional CL differs from the LLM setting.). More recently, De Cao et al. (2021) introduced techniques for editing factual knowledge in language models, marking an early transition toward addressing knowledge updates in the LLM era.

More recent works have adapted continual learning methods to address the specific challenges of LLMs. Huang et al. (2024) introduced Self-Synthesized Rehearsal (SSR), where models generate their own rehearsal data without requiring original training samples, while Elsayed & Mahmood (2024) proposed Utility-based Perturbed Gradient Descent (UPGD) to protect important parameters

while maintaining plasticity through noise injection. Regularization-based methods have also shown promise: Song et al. (2025) combined element-wise and layer-wise importance weighting achieving 20x faster computation, and Li et al. (2024) demonstrated that Sharpness-Aware Minimization flattens loss landscapes to mitigate forgetting. Beyond optimization strategies, Han et al. (2020) proposed biologically-inspired Episodic Memory Activation and Reconsolidation for relation learning, Borhanifard & Faili (2024) combined LoRA with experience replay for dialogue systems, and Sun et al. (2025) used mechanistic interpretability to detect RAG hallucinations. Empirical studies have revealed fundamental insights: Luo et al. (2023) found that larger models forget more but benefit from prior instruction tuning, Kotha et al. (2024) argued forgetting reflects shifted task inference rather than capability loss, and Kalajdzievski (2024) identified precise scaling laws showing forgetting follows an inverse relationship with fine-tuning performance.

These approaches mitigate forgetting but do not analyze how models connect related pieces of information acquired sequentially by studying the episodic binding problem as a function of model scale and narrative complexity. Unlike task-incremental approaches like Huang et al. (2024), we focus on knowledge updates within the same task through continuous addition of individual facts. While we share the use of self-generated rehearsal and parametric knowledge, our goal is understanding how models bind related episodes rather than preventing task interference.

***Most of these works address forgetting in task-incremental continual learning, where models learn distinct tasks sequentially. We study how models connect episodes within a single task, where knowledge is continuously updated through the addition of individual facts and events. Unlike most approaches, we systematically investigate scaling behavior as a function of model scale and narrative complexity, similar to Kalajdzievski (2024) but focused on episodic binding rather than general forgetting. Like Huang et al. (2024), we rely on self-generated rehearsal and parametric knowledge without external retrieval, but we focus on understanding how models bind related episodes rather than preventing interference across tasks.***

### D.3    ADDRESSING CATASTROPHIC FORGETTING THROUGH EDITING

Another research direction attempts to avoid catastrophic forgetting by directly editing specific network components rather than retraining. Fedus et al. (2023) introduced MEMIT, a scalable method for mass-editing factual memories by directly updating MLP weights across multiple layers, successfully editing up to 10,000 memories simultaneously. Das et al. (2024) introduced Larimar, a brain-inspired architecture that couples fast episodic memory with slow semantic memory through a generative pseudo-inverse framework, enabling one-shot knowledge updates without retraining.

***Editing methods target localized factual updates but do not address sequential integration of related episodes, which is the focus of our setting. These techniques were not designed for binding-related challenges where models must connect related episodes learned at different times through shared elements.***

### D.4    DISTINGUISHING PARAMETRIC INTEGRATION FROM RETRIEVAL

While Retrieval-Augmented Generation (RAG) is effective for accessing explicit facts (Lewis et al., 2020), proposing it as a substitute for episodic binding confuses retrieval with the distinct challenge of continual learning. As Lin et al. (2025) argue, RAG provides high-capacity external memory but bypasses the necessity for models to compress and internalize knowledge patterns. We specifically investigate whether models can parametrically bind sequentially learned knowledge, a capability distinct from accessing static buffers.

Furthermore, reliance on retrieval introduces structural dependencies that may obscure the binding problem rather than solve it. Huet et al. (2025) demonstrate that RAG struggles with multi-hop reasoning across episodes, particularly when the chunk size does not strictly align with the episode to be bound, a significant limitation given that optimal chunk sizes are rarely known a priori in real world applications. Similarly, Yoran et al. (2023) and Liu et al. (2024) find that retrieved context can introduce noise that degrades performance on complex inferential chains. Consequently, we exclude non-parametric baselines to isolate the specific bottleneck of parametric integration, ensuring that failures in reasoning are attributable to the model's learning dynamics rather than the confounding variables of retrieval accuracy.

Table 13: Taxonomy of related work across five key dimensions. Our work is the first to jointly address knowledge binding and catastrophic forgetting in a continual learning setting using rehearsal-based methods. Detailed discussion of each work is provided in Appendix D.

| Examples | Knowledge Binding | Catastrophic Forgetting | Scale Effect | Rehearsal | Continual Learning |
|---|---|---|---|---|---|
| Lampinen et al. (2025a) | ✔ | ✘ | ✘ | ✘ | ✘ |
| Treutlein et al. (2024) | ✔ | ✘ | ✘ | ✘ | ✘ |
| Berglund et al. (2023) | ✔ | ✘ | ✘ | ✘ | ✘ |
| Lampinen et al. (2025b) | ✔ | ✘ | ✘ | ✔ | ✘ |
| Allen-Zhu & Li (2024) | ✘ | ✘ | ✘ | ✘ | ✘ |
| Sun et al. (2025) | ✘ | ✔ | ✘ | ✘ | ✘ |
| Kalajdzievski (2024) | ✘ | ✔ | ✔ | ✘ | ✘ |
| Fedus et al. (2023) | ✘ | ✔ | ✘ | ✘ | ✘ |
| Das et al. (2024) | ✘ | ✔ | ✘ | ✘ | ✘ |
| Huang et al. (2024) | ✘ | ✔ | ✘ | ✔ | ✔ |
| Elsayed & Mahmood (2024) | ✘ | ✔ | ✘ | ✘ | ✔ |
| Song et al. (2025) | ✘ | ✔ | ✘ | ✘ | ✔ |
| Li et al. (2024) | ✘ | ✔ | ✘ | ✘ | ✔ |
| Han et al. (2020) | ✘ | ✔ | ✘ | ✘ | ✔ |
| Borhanifard & Faili (2024) | ✘ | ✔ | ✘ | ✘ | ✔ |
| Luo et al. (2023) | ✘ | ✔ | ✘ | ✘ | ✔ |
| Kotha et al. (2024) | ✘ | ✔ | ✘ | ✘ | ✔ |
| Kirkpatrick et al. (2017) | ✘ | ✔ | ✘ | ✘ | ✔ |
| Rebuffi et al. (2017b) | ✘ | ✔ | ✘ | ✔ | ✔ |
| Rusu et al. (2016) | ✘ | ✔ | ✘ | ✘ | ✔ |
| this work | ✔ | ✔ | ✔ | ✔ | ✔ |

*Note: This taxonomy serves as a high-level organizational framework. We acknowledge that individual works may address these dimensions with different emphases or interpretations; our categorization reflects the primary focus of each work as it relates to our research questions.*

## E  LARGE LANGUAGE MODEL USAGE DISCLOSURE

In compliance with ICLR 2026 policies on Large Language Model usage, we disclose the following uses of LLMs:

**Code development and debugging:** Large language models were used as assistants with implementation of the training pipeline, visualization code, data generation and evaluation prompts refinement, as well as plotting utilities. All generated code was reviewed, tested, and validated by the authors before use.

**Writing assistance:** LLMs were also used for rewriting and improving clarity of text passages and the formulation of some technical descriptions. All scientific claims, experimental interpretations, and conclusions remain the original intellectual contribution of the authors.

**Literature review and formulation:** LLMs occasionally assisted in identifying seeds of related work. All referenced works were independently verified by the authors.

The authors take full responsibility for all content in this paper, including any LLM-generated contributions. All experimental results, scientific interpretations, novel insights, and conclusions are the authors' original intellectual work. LLMs served purely as productivity tools and did not contribute to the core research ideas or scientific discoveries presented herein.

## F  BENCHMARK USAGE AND REPRODUCTION GUIDELINES

Here we provide an high level explanation in pseudocode, you can find more details how to reproduce and get exactly the same results inside the code of the repo.

### F.1  HIGH-LEVEL PIPELINE ARCHITECTURE AND DATA FLOW

The benchmark generation pipeline follows a deterministic, multi-stage process designed to create reproducible narrative datasets with controlled complexity. The pipeline consists of five main stages:

**Step 1: Event Generation.** Given fixed parameters (universe, style, number of events $n$, seed), the system first generates atomic narrative events. Each event is sampled from a predefined universe containing temporal markers, entities, spatial locations, and content elements. Events are distributed across the narrative according to a geometric distribution (default parameter for this work $p = 0.3$), which controls the temporal spacing between episodic memories.

**Step 2: Narrative Synthesis.** Events are transformed into coherent narrative chapters through iterative generation using a large language model (Claude 3.5 Sonnet in our experiments). The system performs up to 10 generation attempts per event, implementing verification checks to ensure narrative consistency and proper event grounding. Successfully generated paragraphs are indexed according to the specified book parameters. Valid narrative segments are assembled into a complete book structure. The system builds a ground truth dataframe (`df_book_groundtruth`) that maintains mappings between chapters, original events, and their constituent elements (entities, temporal/spatial markers, content).

**Step 3: Question Generation.** The pipeline automatically generates two types of evaluation questions from the narrative: (i) direct questions testing recall of individual facts within single chapters (SEQ), and (ii) binding questions requiring integration of information across multiple chapters sharing common elements(MEQ). Questions are stratified by complexity (number of chapters involved) and filtered to ensure non-empty answer sets. The system tracks question "widespreadness" to ensure adequate coverage across the narrative timeline. The final stage splits questions into training and evaluation sets, with special handling for single-chapter questions used in fine-tuning. Questions templates are replaced with actual chapter content, and common-sense evaluation data is loaded for baseline comparisons. All artifacts (book, ground truth, questions) are persisted with verification checks to ensure reproducibility across runs.

The complete pipeline maintains deterministic behavior through fixed random seeds and verification checks at each stage, enabling exact reproduction of benchmarks given identical input parameters.

## F.2 Baseline implementation details for SEQ, SEQ+MEQ

All baseline methods share the same fine-tuning data preparation pipeline, differing only in their question selection and generation strategies:

**Static training on SEQ.** This baseline uses only single-chapter questions for fine-tuning. The implementation filters the question dataset to include exclusively questions where `n_chapters_correct_answer == 1`, ensuring that each training example requires information from exactly one narrative chapter. Training data is constructed by pairing each question with its corresponding answer set, formatted as JSON objects with system prompts, user queries, and assistant responses. The final training set is randomly shuffled to prevent learning order biases.

**Static training on SEQ+MEQ.** This variant extends SEQ by including all available questions in the training set, regardless of the number of chapters involved in the answer. The implementation sets `answer_in_one_chapter_only=False`, incorporating both single-chapter questions and multi-chapter binding questions that require integrating information across episodes sharing common elements. This provides explicit supervision for learning cross-episode relationships during sequential fine-tuning.

**Sequential Training.** This baseline implements continual learning by training on individual narrative chapters sequentially, one at a time, in their temporal order. During training, the model is fine-tuned on the content of each chapter independently. After processing all chapters in sequence, the model is evaluated on questions whose answers require information distributed across multiple chapters from the learned sequence. This approach tests whether the model can spontaneously bind related episodes through their shared elements, representing a pure continual learning scenario where cross-episode integration emerges only at test time.

All methods utilize the same JSON formatting pipeline, where questions are paired with their ground-truth answer sets and wrapped with consistent system prompts instructing the model to provide accurate, complete responses based on the narrative context. The training data undergoes random shuffling before fine-tuning to ensure robust learning across all episodic positions.

### F.3 STEP-BY-STEP EVALUATION PROTOCOLS

To ensure fair and objective comparison across all training approaches, we employ a standardized evaluation protocol with consistent inference and assessment procedures:

**Step 1: Model Loading.** For each training method, we load the fine-tuned model checkpoint. The system supports both LoRA adapters and fully fine-tuned models, automatically detecting the fine-tuning type and loading from the appropriate checkpoint path. All models use the same base architecture (e.g., Llama-3-8B) with identical tokenizer configurations and maximum sequence length of 8192 tokens.

**Step 2: Inference Generation.** Each model generates answers for all test questions using identical inference parameters: temperature = 0.1 for controlled generation, consistent system prompts providing task instructions, and uniform maximum token limits for responses. The inference pipeline processes questions sequentially, storing both generated answers and any reasoning traces for subsequent analysis.

**Step 3: LLM-as-Judge Evaluation.** To assess answer quality beyond exact match metrics, we employ a consistent LLM judge (detailed of the prompt in Appendix A.2). The judge model evaluates all generated responses using the same prompt template and temperature setting (0.1), ensuring objective comparison across methods A.4. This approach evaluates semantic correctness, completeness, and relevance of generated answers against ground truth, providing robust assessment even when models produce valid paraphrases or alternative phrasings.

**Step 4: Metrics Computation.** For each model, we compute both exact match accuracy and LLM-judged correctness scores, stratified by question type (single vs. multi-chapter) and complexity. Results are aggregated across multiple random seeds to ensure statistical significance of performance differences.

**Note on Evaluation Design.** The test set construction allows for partial overlap with training questions, particularly for multi-episode questions (MEQs), though the test and training sets are not identical due to different sampling seeds and stratification procedures. This design enables us to evaluate whether models can effectively learn and retrieve episodic bindings that were explicitly presented during training, which is essential for assessing the core binding capability rather than testing generalization to entirely unseen question patterns.

This unified evaluation framework ensures that observed performance differences stem from the training methodologies rather than inference artifacts or evaluation inconsistencies.

