# OpenReview forum: "Episodic Knowledge Binding: a New Challenge for LLM Continual Learning"
_ICLR.cc/2026/Conference — ICLR 2026 Conference Desk Rejected Submission_

### Official Review · Reviewer_8HcV · 2025-10-25

**Soundness:** 2
**Presentation:** 1
**Contribution:** 1
**Rating:** 2
**Confidence:** 3

**Summary:**

This paper aims to address the episodic knowledge binding problem in LLMs; that is, enhancing their ability to interlink and recall multiple related events, each defined by a tuple of time, space, entity, and content. The authors generate a series of synthetic episodic events and evaluate model performance on memory recall using a lenient recall metric. They find that models like Llama and GPT-4.1 perform poorly on this task. To improve performance, they employ Supervised Fine-Tuning (SFT) with a key innovation: for each new event, the model first generates related past events (e.g., those sharing the same entity or time). These related events are then merged with the current event to create the SFT training data, rather than using the raw event alone. The authors demonstrate that this sophisticated method for generating training data leads to improvements in episodic memory recall.

**Strengths:**

The paper takes on an important question regarding binding problems for episodic events in LLMs. The authors show that augmenting simple SFT with a more sophisticated, relation-aware method for organizing training data effectively enhances the model's episodic memory capabilities.

**Weaknesses:**

The manuscript would benefit from significant revisions to improve its clarity and structure, as several typos and an ineffective narrative flow currently hinder readability (see minor comments for specific examples).

While I appreciate the effort to categorize the related literature in Table 2, its utility is limited without clear, commonly-agreed-upon definitions for the column headers (e.g., "knowledge binding," "rehearsal," "continual learning"). This makes it difficult to assess the accuracy of the taxonomy.

The core methodological proposal for enhancing episodic memory involves querying the model's own parametric memory to retrieve events related to a new episodic event *k*. These recalled events are then synthesized with event *k* to create the training data for SFT. A more fundamental concern is the paper's heavily engineering-focused presentation, which lacks scientific insight into why the proposed method should work. The approach remains a form of SFT, which compresses episodic information into the model's parametric memory. If the fundamental limitations of parametric memory (e.g., binding failures, catastrophic forgetting) are the root cause of the binding problem, it is unclear why a solution that ultimately relies on further compressing information into that same parametric memory would resolve these issues. The manuscript would be significantly strengthened by a deeper theoretical discussion of this apparent paradox.

Minor comments:
Line 136: "e.g..," --> "e.g.,"
Line 323: in "structure. 7", "7" should be removed?
Line 399: in Figure 2's caption "List all times Emma was in Tokyo" --> "List all times Mary was in Tokyo"? The figure shows Mary (colored in red), but in the main text the running example was Emma. This is very confusing.

**Questions:**

Another key concern of the proposed method is the risk for model collapse or instability. The training data consists of a combination of the model's own recollections and the new event. Why this self-referential training data generation, where a significant volume of synthetic data is compressed into the weights via SFT, does not lead to degenerative feedback or overfitting?

---

> ### Author Response · Authors · 2025-11-24
> **Clarification: binding as an indexing failure, not a capacity limitation**
>
> > The paper takes on an important question regarding binding problems for episodic events in LLMs. The authors show that augmenting simple SFT with a more sophisticated, relation-aware method for organizing training data effectively enhances the model's episodic memory capabilities.
>
> Thank you for acknowledging the methodological proposal of GCR. We would like to add that the primary scientific contribution is diagnostic: we isolate and formalize Episodic Knowledge Binding as a distinct failure mode in LLMs (a failure that persists even when catastrophic forgetting is eliminated through static training (Figure 1). We hence respectfully push back on the characterization of our work as "engineering-focused" lacking scientific insight. Our work is a falsifiable, reproducible finding about the fundamental organization of parametric memory. GCR then serves as existence proof that the problem is addressable, not as the definitive solution. Dismissing diagnostic work because the initial remedy is "engineered" conflates two distinct contributions.
>
> We ask the reviewer to evaluate the diagnostic contribution on its own merits, independent of the solution's maturity.
>
> > The core methodological proposal for enhancing episodic memory involves querying the model's own parametric memory to retrieve events related to a new episodic event k. These recalled events are then synthesized with event k to create the training data for SFT. A more fundamental concern is the paper's heavily engineering-focused presentation, which lacks scientific insight into why the proposed method should work. The approach remains a form of SFT, which compresses episodic information into the model's parametric memory. If the fundamental limitations of parametric memory (e.g., binding failures, catastrophic forgetting) are the root cause of the binding problem, it is unclear why a solution that ultimately relies on further compressing information into that same parametric memory would resolve these issues. The manuscript would be significantly strengthened by a deeper theoretical discussion of this apparent paradox.
>
> The reviewer frames this as a paradox, but we argue the apparent paradox is due to the conflation of two phenomena. Binding failure is not a failure of parametric capacity, it is instead a failure of parametric "indexing". Our results demonstrates this directly: models achieve high SEQ accuracy (the facts are stored) but low MEQ accuracy (the facts are not cross-linked). The base information exists in the weights; what is missing are the associative pathways to retrieve it exhaustively.
>
> GCR addresses this by restructuring how information is organized during encoding, not by adding more storage. The biological analogy is memory reconsolidation: when animals face new information that relates to existing memory representations, those old traces become *labile* and hence ready to absorb new associations. During this process, animals also do cued rehearsal (recall related memories).
> GCR simulates this by activating related traces before consolidation. Since transformers lack an intrinsic mechanism for spontaneous cued-recall, external cued retrieval is architecturally necessary (not an engineering shortcut, but a required component given current architectures).
>
> So this is not "compressing more into the same memory"; it is reorganizing the index structure of that memory.
>
> Finally, we would like to not that our work deliberately disentangles the components of this problem:
> - By using single-shot training, we isolate binding from catastrophic forgetting and hallucination. We show that even if those issues were completely solved, binding failures would persist.
> - Our various GCR-Filtered results demonstrate that even when hallucinations are removed, binding and catastrophic forgetting remain challenging problems.

---

> > ### Author Response · Authors · 2025-11-24
> > **Mitigating factors against degenerative feedback**
> >
> > > Another key concern of the proposed method is the risk for model collapse or instability. The training data consists of a combination of the model's own recollections and the new event. Why this self-referential training data generation, where a significant volume of synthetic data is compressed into the weights via SFT, does not lead to degenerative feedback or overfitting?
> >
> > We acknowledge the risk but note three mitigating factors and one empirical evidence against degenerative feedback.
> >
> > First, (i) GCR is not blind replay, since the Merger module performs a structured reorganization of information, updating the associative index rather than amplifying existing weights.
> >
> > Second, (ii) our GCR-Filtered results demonstrate that when hallucinations are controlled, binding improves substantially. This localizes the instability to hallucination (an orthogonal, solvable problem) rather than the binding mechanism itself.
> >
> > Third, the training signal contains ground-truth anchoring: the new event $E_k$ provides veridical information that constrains the synthetic MEQs. Degenerative feedback requires error to compound without correction; when hallucinations are controlled, GCR's structure prevents this through the veridical anchor of each new episode.
> >
> > Fourth, empirically, GCR improves binding even with weak 8B models that have limited parametric capacity and high hallucination rates (Figure 2). If self-referential training caused degenerative collapse, we would observe performance degradation over the training sequence; instead, GCR is still doing better compared to classic continual learning (note that we have also included O-LoRA now in our revision, a successful state of the art continual learning method).
> >
> > > While I appreciate the effort to categorize the related literature in Table 2, its utility is limited without clear, commonly-agreed-upon definitions for the column headers (e.g., "knowledge binding," "rehearsal," "continual learning"). This makes it difficult to assess the accuracy of the taxonomy.
> >
> > Fair point. We moved the table to the appendix, added column definitions and a note that our categorization reflects each work's primary focus relative to our research questions. Since this was also asked by another reviewer, we rewrote a sharper and hopefully more effective related work section in the main text.

---

> ### Author Response · Authors · 2025-11-24
> **Why we ask for reconsideration**
>
> Summarizing, we ask the reviewer to evaluate this work primarily as a diagnostic contribution: does the binding phenomenon we isolate represent a genuine, previously uncharacterized limitation of LLMs? We believe our controlled experiments demonstrate that it does.
>
> In our revision, we reworked the presentation, sharpened the link to the related work illustrating while this novel problem of binding emerged with LLMs but not with classic continual learning research, and added new experiments (a new recent SoTA continual learning baseline, experiments to vary the number of epochs and verify whether a form of "grokking" can "appear" to solve binding, extension to 70B model size, and additional experiments on the (un)learnability of binding that we'll be adding shortly). We believe the revised manuscript now clearly separates the diagnostic contribution (novel, reproducible) from the methodological contribution (initial, improvable), and merits reconsideration on both fronts.

---

### Official Review · Reviewer_CpjG · 2025-10-31

**Soundness:** 3
**Presentation:** 2
**Contribution:** 3
**Rating:** 6
**Confidence:** 3

**Summary:**

This paper identifies **Episodic Knowledge Binding** as a fundamental limitation in large language models: their inability to retrieve multiple related episodes through shared elements when trained only on individual, single-event examples. Unlike humans who naturally connect separate experiences (e.g., recalling all encounters with a person), LLMs fail at exhaustive multi-event retrieval despite excelling at learning isolated facts.
The authors distinguish this from catastrophic forgetting, showing the binding failure persists even with aggregated training data. Through controlled experiments using synthetic episodic narratives, they demonstrate a consistent **binding gap** across model scales (3B–13B parameters and GPT-4.1): models achieve high accuracy for single-event queries but experience severe performance collapse for multi-event retrieval. Surprisingly, model scaling offers minimal improvement within the tested range.
To address this, the paper proposes **Generative Cued Replay (GCR)**, inspired by hippocampal memory consolidation. When processing new episodes, GCR queries the model for related prior episodes and synthesizes multi-event training data without storing past episodes. This approach improves binding without architectural changes, offering a practical alternative to computationally infeasible exhaustive multi-event supervision.
The authors release an Episodic Knowledge Binding benchmark to facilitate future research on this capability gap in current LLMs.

**Strengths:**

- **Novel Problem Formulation**: The paper identifies and formalizes Episodic Knowledge Binding as a distinct challenge from catastrophic forgetting, addressing a fundamental gap between human and LLM memory systems that has been largely overlooked in prior work.
- **Comprehensive Empirical Analysis**: The study systematically evaluates the binding gap across multiple model scales (3B-13B parameters and GPT-4.1) and narrative complexities (10-100 events), demonstrating consistency of the phenomenon and providing important insights about scaling limitations.
- **Clear Distinction from Existing Problems**: The authors carefully distinguish episodic binding from catastrophic forgetting by showing the problem persists even with aggregated (non-sequential) training data, establishing this as a fundamental limitation rather than a continual learning artifact.
- **Biologically-Inspired Solution**: Generative Cued Replay (GCR) draws meaningful inspiration from hippocampal memory consolidation, providing both theoretical grounding and practical effectiveness without requiring architectural modifications.
- **Practical Applicability**: The proposed GCR method addresses a real scalability challenge—the computational infeasibility of exhaustive multi-event supervision—while achieving significant improvements through synthetic data generation from the model's own parametric memory.
- **Benchmark Release**: The authors provide an Episodic Knowledge Binding benchmark to facilitate reproducibility and future research on this important capability gap.
- **Well-Motivated Research Question**: The paper addresses a cognitively fundamental capability that is essential for practical applications requiring comprehensive information retrieval, making the work both theoretically interesting and practically relevant.

**Weaknesses:**

**1. Insufficient Support for "Binding is Learnable" Claim**

The paper claims that Train(SEQ+MEQ) "proves binding is learnable," but this conclusion is not adequately supported. Train(SEQ+MEQ) merely demonstrates that models can memorize multi-event QA pairs when explicitly supervised and produce correct output formats—an unsurprising result of direct supervision. Critically, it does NOT demonstrate that genuine "binding" (flexible association and retrieval of related episodes) has been learned, nor that models can generalize to unseen multi-event queries.

The distinction between memorization and binding is fundamental to the paper's thesis. To substantiate the "learnable" claim, the authors must demonstrate: (1) generalization to held-out multi-event queries not seen during training, (2) transfer to different types of binding relationships, and (3) evidence that models develop flexible retrieval mechanisms rather than task-specific lookup tables. The paper should include experiments testing generalization on held-out multi-event queries, analysis of binding novel entity-attribute combinations, and systematic comparison of binding quality versus mere task completion.

**2. Inadequate Figure Documentation**

Figure 1 lacks sufficient explanation. The meaning of numbers in parentheses is unclear, and "narrative types" are not defined or explained in the caption or main text, making the figure difficult to interpret without extensive cross-referencing.

**3. Inconsistent Terminology**

The paper inconsistently uses "accuracy" and "recall" throughout, creating confusion about which metric is being reported and whether they are being used interchangeably or represent different measurements.

**4. Poor Paper Organization**

The paper suffers from structural issues that impede readability and comprehension:

- **Missing Related Work Context**: Important baseline methods should be explained in the related work section with explicit comparisons to the proposed GCR method. Readers need to understand how GCR differs from and improves upon existing approaches.

- **Misplaced Implementation Details**: Section 3.1 contains implementation details that would be better suited for the appendix, cluttering the main narrative with technical minutiae.

A restructuring that moves technical details to appendices while expanding related work comparisons and benchmark documentation in the main text would significantly improve the paper's clarity and impact.



**5. Non-Deterministic Evaluation Methodology**

The paper's reliance on "LLM as a judge" for extracting model predictions introduces indeterminacy into the evaluation process. While this approach may offer flexibility in parsing varied response formats, it creates several concerns:

- **Reproducibility**: Different LLM judge implementations, versions, or even sampling parameters could yield different recall scores for identical model outputs, making results difficult to reproduce across different research groups.

- **Evaluation Validity**: The LLM judge itself may have parsing errors, biases, or inconsistencies that confound the measurement of the actual binding capability being studied.

- **Transparency**: Without deterministic evaluation, it becomes difficult to diagnose whether performance differences stem from genuine binding improvements or variations in the judge's extraction quality.

**Questions:**

1. **Regarding the "binding is learnable" claim**: How do you distinguish between memorization of specific multi-event QA pairs and genuine binding capability in your Train(SEQ+MEQ) experiments? Can you provide evidence of generalization to held-out multi-event queries that were not seen during training?

2. **On evaluation of binding vs. task completion**: Have you tested whether models trained with Train(SEQ+MEQ) can bind novel entity-attribute combinations that differ from training examples? What experiments demonstrate that flexible retrieval mechanisms are learned rather than task-specific lookup tables?

3. **Clarification on Figure 1**: What do the numbers in parentheses represent in Figure 1? Can you define and explain what "narrative types" means in the context of your experiments?

4. **Metric consistency**: The paper uses both "accuracy" and "recall" throughout. Are these terms being used interchangeably, or do they represent different measurements? Can you clarify which metric is reported in each section and provide consistent definitions?

5. **LLM-as-judge reproducibility**: Given that your evaluation relies on an LLM judge to extract model predictions, how do you ensure reproducibility across different implementations, LLM versions, or sampling parameters? Have you conducted any analysis on the inter-rater reliability or consistency of the LLM judge?

6. **Deterministic evaluation baseline**: Would it be possible to include a deterministic evaluation setting (e.g., structured output formats with rule-based parsing) alongside the LLM judge to provide a reproducible baseline for the benchmark?

7. **Benchmark documentation**: Can you provide more details about the benchmark in the main paper, including baseline method comparisons, comprehensive dataset statistics, and representative samples? What are the specific evaluation protocols that other researchers should follow?

8. **Related work discussion**: The related work section currently only contains a taxonomy table (Table 2) with comprehensive discussion deferred to Appendix E. This is insufficient for the main paper. Can you provide a proper related work discussion in the main text that explains key prior methods (e.g., experience replay, elastic weight consolidation, memory-augmented approaches) and explicitly positions your work relative to these techniques? The table could be condensed or moved to the appendix to make room for this critical discussion.

---

> ### Author Response · Authors · 2025-11-24
> **Scope clarification: is binding learnable? memorization vs generalization**
>
> Thank you for recognizing the strengths of our work. Please find some clarifications and answers below.
>
> > 1. Insufficient Support for "Binding is Learnable" Claim
> > Regarding the "binding is learnable" claim: How do you distinguish between memorization of specific multi-event QA pairs and genuine binding capability in your Train(SEQ+MEQ) experiments? Can you provide evidence of generalization to held-out multi-event queries that were not seen during training?
>
> **Short answer**
> Thank you for giving us the chance to clarify an apparent ambiguity. When we (*unfortunately*) wrote "multi-event supervision proves binding is learnable," we meant that models have sufficient parametric capacity to store and retrieve bound episodes. Our central finding, however, is that this capacity is never spontaneously activated by single-event training. The distinction matters: binding requires explicit multi-event supervision to emerge, which is precisely why GCR synthesizes such supervision on-the-fly. We have revised the paper to make this distinction explicit.
> Finally, your ask about generalization vs memorization is spot-on for future communications about our work: this paper is about episodic **memory** (one of the two families of declarative memory, the other being semantic memory). So it is about (better) memorization, not generalization.
>
> **Longer answer**
>
> **why good memorization is still needed** LLMs are bringing new problems compared to classic ML, episodic knowledge binding is one of them, **which this paper is the first to diagnose and study systematically.**  Unlike classic ML where the focus is on generalization, we want LLMs to also memorize salient events about our world. So this paper is not about generalization, but better-quality memories.
>
> **The binding challenge is about seen, not unseen events.** All events in our experiments are seen during training : this is a deliberate choice that is essential to our design:
>
> - We train models on a narrative where they learn each event individually (e.g., "Mary was in Tokyo on March 5th", "Mary was in Tokyo on Sept 12th")
> - We then test if they can answer "When was Mary in Tokyo?" → "March 5th, Sept 12th"
> - **The model has already seen both events during training**  the challenge is whether it can retrieve them together
>
> > "Can you provide evidence of generalization to held-out multi-event queries?"
>
> Given the clarification above, we believe it is clear for all of us now that answering multi-event queries about single events never seen during training is not reasonable (e.g. asking someone to list all times they visited a city they've never been to : not a binding failure but complete absence of information).
>
> he other remaining possibility (learning the binding pattern from some entities and "applying" it to others) is reasonable, but unsuccessful and most of all : impractical. To illustrate concretely: if training on MEQ(John, Sarah) helped the model answer MEQ(Mary), that would suggest binding is a transferable skill. We tested this systematically across narrative universes (we did not put these negative results initially, but we now added them to our revision in appendix C.1.2). Specifically, we trained on SEQ for 2 universes (sci-fi, world news) plus MEQ supervision for just 1 (news), then evaluated SEQ and MEQ performance on the sci-fi universe. Despite the apparent improvement at 500 epochs, we show in Appendix C.1.1 that extended training beyond degrades the model's general abilities without producing binding capability. Results: MEQ performance on the "unsupervised universe" remained low despite the model having (i) memorized all its single events and (ii) received explicit binding supervision on another structurally identical narrative (news). We have no reason to believe that binding transfers, and our first approach is that it must be induced for each episodic cluster, which is why GCR's online synthesis is necessary.
>
> Finally, multi event supervision, even if it is partial is impractical for real-world scenarios where multi-event questions are rarely available in training data. Imagine the current training data batch contains single events that are related to past training data, how can we find the past related multi event questions to force the binding "generalization"?  We believe that the only known way to generate them at scale is through synthetic generation, which is what approaches like Entigraph do for static training (but need building graph for the whole training data), and this is precisely what GCR addresses in an online manner (without need the full training data).
>
> So to summarize, the binding problem is a memorization problem, and stands alone as interesting, even in the absence of classic ML train/test splits. Even when all events are seen during training, binding is desirable and challenging, and for which there exists no solution today. To the best of our knowledge, Generative Cued Recall is the first baseline in this space.

---

> ### Author Response · Authors · 2025-11-24
> **Addressing remaining points**
>
> > Inadequate Figure Documentation
> > 3. Clarification on Figure 1
>
> Thank you. We updated the figure description to clarify.
>
> > Inconsistent Terminology [...] inconsistently uses "accuracy" and "recall" throughout
>
> Thank you for highlighting this inconsistency. We reviewed the manuscript and standardized our terminology throughout. All reported metrics in the paper use the lenient recall evaluation framework detailed in Sec 2.2 (Question types and evaluation framework), where we consider an answer correct if it includes the expected information, regardless of additional content.
>
> > Poor Paper Organization [..]
> > 9. Related work discussion: The related work section currently only contains a taxonomy table (Table 2) with comprehensive discussion deferred to Appendix E. This is insufficient for the main paper. Can you provide a proper related work discussion in the main text that explains key prior methods (e.g., experience replay, elastic weight consolidation, memory-augmented approaches) and explicitly positions your work relative to these techniques? The table could be condensed or moved to the appendix to make room for this critical discussion
>
> Thank you. We completely restructured the related work presentation to sharpen the novely of the binding problem and its relation to the literature, please have a look at the revised related work section, which we expaned. We also better clarified the extended version in the appendix D as you suggested.
>
> > Non-Deterministic Evaluation Methodology
> > 5. LLM-as-judge reproducibility
> > 6. Deterministic evaluation baseline
>
> We appreciate the reviewers' concerns about reproducibility. In this paper, we reused the evaluation methods developed by [Huet et al., ICLR 2025]. As noted in that work, the use of LLMs is mechanical rather than subjective since the LLM's job is to simply extract factual items from responses, which are later used for deterministic comparison against ground truth, hence not making any qualitative judgments. That being said, to further address these concerns, we have included Appendix A.4 and Figure 5 showing a comparison between 3 state of the art llms and a human validation on 100 examples demonstrating near-perfect agreement.
> | Model | Agreement (%) |
> |:------|:-------------:|
> | GPT-4o-mini | 98 |
> | Claude Haiku 4.5 | 99 |
> | Gemini 2.0 Flash | 99 |
> | Human Expert | 100 |
>
> All LLM judges achieve 98-99% agreement with human expert annotations.
> > 7. Benchmark documentation: Can you provide more details about the benchmark in the main paper, including baseline method comparisons, comprehensive dataset statistics, and representative samples? What are the specific evaluation protocols that other researchers should follow?
>
> Thank you for the suggestion. While we provided examples of event generation in Appendix A.1 and implementation details in Appendix A.5 , we agree that clearer benchmark documentation would benefit reproducibility. We added a new appendix F  that will include:
> (1) High-level pipeline architecture and data flow
> (2) Baseline implementation details for SEQ, SEQ+MEQ
> (3) Step-by-step evaluation protocols
> The accompanying repository will contain exact commands for reproduction, pre-generated datasets, and evaluation scripts. This will ensure other researchers can easily adopt our benchmark and fairly compare future methods.
>
> Thank you again.

---

### Official Review · Reviewer_xKvt · 2025-10-31

**Soundness:** 2
**Presentation:** 2
**Contribution:** 2
**Rating:** 2
**Confidence:** 4

**Summary:**

The paper defines Episodic Knowledge Binding (EKB): if training only on single-event QA pairs, can an LLM retrieve all related episodes matching a cue (multi-event QA) from parametric memory? The paper creates synthetic narratives with tuples (t, s, ent, c) as training data and shows that models excel on single-event queries but collapse on multi-event retrieval, and the gap widens with narrative length and is not fixed by scale. They propose Generative Cued Replay (GCR): when a new event arrives, query the current model for related prior episodes and merge them with the new event to synthesize multi-event supervision online. GCR is shown to improve multi-event binding vs. continual single-event fine-tuning, with strongest gains when hallucination filtering is applied.

**Strengths:**

- The problem statement is clear, highlighting an important problem of learning the relations across multiple events.
- Evaluation spans two model families and multiple sizes (Llama-3 3B/8B/13B and GPT-4.1 variants), and multiple narrative lengths (10/30/100), supporting generality claims.
- The writing is clear and easy to follow.

**Weaknesses:**

- GCR could use more training compute than other baseline methods. When compared to baselines like single-event fine-tuning etc., it is not clear if the number of FLOPs or tokens are controlled.
- Missing Baselines: RAG/external memory is dismissed based on prior work; there is no RAG or memory module baseline. In addition, prior work has shown that rephrasing or other data augmentation techniques can improve down-stream task performance. These baselines are not explored at all.
- Evaluation uses an LLM judge with lenient recall and counts an answer correct only at recall = 1.0; the paper does not report human audit or inter-annotator checks for this metric, leaving reliability uncertain.
- Figure 3 is over-stretched vertically and hard to read.

**Questions:**

1. Can you report FLOPs or total tokens processed for each training condition (SEQ, SEQ+MEQ upper bound, and each GCR variant)? Are these budgets comparable across methods?
2. What error analysis have you done for different training methods (SEQ, SEQ+MEQ, and GCR)? How do these failure cases reveal about the shortcomings of each method?
3. With a multi-turn RAG baseline, how does performance compare to GCR on episodic binding? Is this task non-trivial for retrieval-only methods?

---

> ### Author Response · Authors · 2025-11-24
> **Scope clarification (parametric memory and continual learning) and new experiment on training budgets**
>
> > Strengths [..] The problem statement is clear, highlighting an important problem of learning the relations across multiple events.
> > The writing is clear and easy to follow.
>
> Thank you for your service. Please find our answers below.
>
> > [Weaknesses] GCR could use more training compute than other baseline methods. When compared to baselines like single-event fine-tuning etc., it is not clear if the number of FLOPs or tokens are controlled.
> > Can you report FLOPs or total tokens processed for each training condition (SEQ, SEQ+MEQ upper bound, and each GCR variant)? Are these budgets comparable across methods?
>
> Thank you for the question. We want to first clarify that comparing GCR directly with single-event fine-tuning in terms of computational cost would miss the main purpose of our experimental design. SEQ, SEQ+MEQ upper bound, and each GCR variant are not "competitors" but rather controlled experimental conditions.
>
> For example, in the static (non-continual learning) setting, training on (i) single-event questions (SEQ) and (ii) multi-event questions (MEQ) serve as a diagnostic tool to demonstrate that the episodic binding weakness exists even in the simplest scenario, where catastrophic forgetting and hallucination do not come into play.
> - What SEQ experiment shows: Models that learn all single events (e.g. know all the locations of Mary, one by one) fail to answer multi event questions (i.e. fail to list all the locations where Mary was).
> - What training on (SEQ+MEQ) simply shows: models have enough capacity to learn the multi event questions, so it's not a matter of model size. So training on (SEQ+MEQ) is just a "naive control experiment". The most important experiment here is the failure of SEQ.
>
> Nonetheless, not reported initially, we had checked whether something like "grokking" could help binding in the case of Single-event training: we let the the model train on single event questions for as long as 3000 epochs, with no success. We added this experiment in Fig.8 and detailed it in Appendix C.1. Even with this massive increase in training time, multi-event performance remains persistently low while single-event accuracy stays high. While MEQ improves at 3k epochs, this reflects brute-force memorization. The CS (commonsense) column shows significant knowledge loss, confirming the model sacrifices general capabilities for task overfitting.
> | Epoch | SEQ | MEQ | CS |
> |------:|----:|----:|---:|
> | 0 | 0.0 | 0.0 | 72.3 |
> | 1 | 0.0 | 0.0 | 71.5 |
> | 5 | 36.9 | 15.8 | 71.0 |
> | 10 | 40.0 | 21.6 | 70.9 |
> | 15 | 57.7 | 21.2 | 70.8 |
> | 20 | 82.3 | 19.8 | 70.7 |
> | 40 | 86.7 | 17.1 | 70.6 |
> | 50 | 88.5 | 19.8 | 70.7 |
> | 500 | 95.4 | 26.6 | 71.0 |
> | 3000 | 98.5 | 36.5 | 63.8 |
>
> The same applies to GCR, all the GCR variants are controlled experiments rather than competitors that could achieve the same performance as GCR if they were given more computational budget. **To the best of our knowledge, there exists no solution in the literature today to the binding problem. So we believe that at this stage, any baseline that works is a good starting point, on which future work could build.**
>
>
> > Missing Baselines: RAG/external memory is dismissed based on prior work; there is no RAG or memory module baseline. In addition, prior work has shown that rephrasing or other data augmentation techniques can improve down-stream task performance. These baselines are not explored at all.
> > With a multi-turn RAG baseline, how does performance compare to GCR on episodic binding? Is this task non-trivial for retrieval-only methods?
>
> We would like to first emphasize that this is a continual learning research paper that focuses on parametric memory (as also mentioned in the title of the paper). While RAG can mitigate some symptoms, it does not address the fundamental questions: can neural networks learn to compose knowledge, or are they limited to pattern matching on memorized examples? This has implications for continual learning, reasoning capabilities, and whether scaled-up versions of current architectures can achieve robust generalization.
>
> In this work, we designed a benchmark for continual learning, adapted for small ~7B models. Our largest dataset of 100 events fits within a single context window. This means that not only RAG-based solutions (if they are effective in multi-hop retrieval) can trivially solve it, but even in-context learning could theoretically solve this instance of the binding problem.
>
> Finally, while we acknowledge the utility of the "philosophical" debate of whether human-like continual learning will be needed or whether RAG will solve all problems, we believe it should not be a reason to dismiss research on parametric memory and continual learning. We rewrote Sec2.4 which might have been misleading to sharpen the scope of our work.
>
> Our work is the first to highlight a new failure mode of parametric memory in LLMs and we believe that this contributions was not weighted in in the evaluation of our work.

---

> ### Author Response · Authors · 2025-11-24
> **Addressing remaining points**
>
> > Evaluation uses an LLM judge with lenient recall and counts an answer correct only at recall = 1.0; the paper does not report human audit or inter-annotator checks for this metric, leaving reliability uncertain.
>
>  We appreciate the concerns about reproducibility. In this paper, we reuse the evaluation methods developed by [Huet et al., ICLR 2025]. As noted in that work, the use of LLMs is mechanical rather than subjective since the LLM's job is to simply extract factual items from responses, which are later used for deterministic comparison against ground truth, hence not making any qualitative judgments. That being said, to further address these concerns, we have included Appendix A.4 and Figure 5 showing a comparison between 3 state of the art llms and a human validation on 100 examples demonstrating near-perfect agreement.
>   | Model | Agreement (%) |
> |:------|:-------------:|
> | GPT-4o-mini | 98 |
> | Claude Haiku 4.5 | 99 |
> | Gemini 2.0 Flash | 99 |
> | Human Expert | 100 |
>
> All LLM judges achieve 98-99% agreement with human expert annotations.
> > Figure 3 is over-stretched vertically and hard to read.
>
> Fixed, thank you, this was due to a last minute wrong manipulation before submission.
>
> > What error analysis have you done for different training methods [...]? How [...] method?
>
> We interpret this as asking whether different training conditions produce qualitatively distinct failure modes. If so, they do as SEQ-trained models fail by returning only a single event (e.g. most recent seen in training), while GCR-trained models return partial lists. Another failure mode that is frequent with such small models is hallucinations which we control in our experiments.

---

> ### Comment · Reviewer_xKvt · 2025-11-27
> **Responses to the Authors**
>
> Thanks for the update. We do not agree with the claim that "this work is the first to identify a new failure mode of parametric memory in LLMs." In fact,  one of the works cited by the authors (Huet et al., 2025) has already conducted similar experiments. While this paper differs in using single-sentence events instead of the multi-paragraph synthetic chapters used in earlier benchmarks, this design choice is rather incremental and does not introduce a fundamentally new phenomenon. The main findings here are also not surprising, as prior work has already shown that for fine-tuning baselines, performance decreases when moving from single-event to multi-event retrieval. As such, the novelty of the contribution appears very limited, and we choose to maintain our original score.
>
> References:
> - Huet, Alexis, Zied Ben Houidi, and Dario Rossi. "Episodic Memories Generation and Evaluation Benchmark for Large Language Models." The Thirteenth International Conference on Learning Representations.

---

> > ### Author Response · Authors · 2025-11-28
> >
> > We respectfully disagree with the characterization of our contribution relative to Huet et al. (2025):
> > - First, the fine-tuning experiments in Huet et al. are *preliminary and anecdotal, as the authors themselves acknowledge*, explicitly stating in their conclusion: "However,we acknowledge limitations that open avenues for future research: […] The observed performance limitations suggest that current fine-tuning methodologies may not be optimally suited for episodic memory tasks, underscoring the need for developing new strategies for the broad scientific community." **Their setup consists of a single run experiment via OpenAI API calls with one seed**. In contrast, we systematically study binding failures across model scales and fine-tuning methods; including LoRA, full fine-tuning, and OLoRA across different configurations (total 5k hours of training)
> >
> > - Second, Despite its importance the problem is understudied. Recent unpublished preprints (1 month ago) from Google DeepMind (https://arxiv.org/pdf/2509.16189, https://arxiv.org/pdf/2505.0066) motivate similar parametric memory limitations and conceptually explore new approaches to address them. These works (conceptually motivate the problem but do not provide solution to measure nor solve it). They leave the discussion of how to address this problem in CL scenarios essentially unexplored, which is precisely the gap our work fills through systematic experimentation and our GCR pipeline. Important and not visible enough.
> >
> > As a summary, we took Huet et al.’s conclusion as a source of inspiration (which we cite), directly addressing the gap they mention in their conclusions; by providing the systematic empirical investigation and new approaches they suggested were necessary. Beyond identifying the binding failure in current LLMs, our contributions include (1) systematic evidence across model scales (3B to 70B parameters), showcasing that this is not simply a data scale issue but a fundamental limitation of current training paradigms and finally (2) introduction and evaluation of Generative Cued Replay (GCR) as a human inspired concrete solution (with corresponding ablations).

---

### Author Response · Authors · 2025-12-04

# Response to Area Chair

## Why This Paper Deserves Acceptance

When an LLM learns "Mary visited Tokyo" in one training session and "Mary visited London" in another, can it later answer "Which cities did Mary visit?" We demonstrate that it cannot, achieving only 12.4% accuracy on simple binding queries and 0% on complex ones, despite models from 3B to 70B parameters successfully recalling each fact in isolation.

This is not catastrophic forgetting (the facts are retained) nor a reasoning failure (the inference is trivial). It is a fundamental gap in how parametric memory represents knowledge acquired across time, a gap we systematically characterize for the first time.

We present 5,000+ training hours of controlled experiments spanning multiple model families (LLaMA, GPT), scales (3B–70B), and narrative complexities (10–100 events). We release a benchmark for reproducibility and propose Generative Cued Replay (GCR), a biologically-inspired method that nearly doubles performance on simple queries and enables non-zero performance on previously impossible complex bindings.

The timeliness of this problem is independently reinforced by concurrent work from Google DeepMind published after our submission: Lampinen et al. (2025a) show that models fail at "latent learning" and flexible reuse of experiences, while Lampinen et al. (2025b) reveal that fine-tuning and in-context learning generalize in fundamentally different ways. These works conceptually motivate the problem but do not provide solutions for continual learning scenarios. This is precisely the gap our work fills through systematic experimentation and our GCR pipeline.

**References:**
- Lampinen et al. (2025a) *On the generalization of language models from in-context learning and finetuning: a controlled study*. arXiv:2505.00661
- Lampinen et al. (2025b) *Latent learning: episodic memory complements parametric learning by enabling flexible reuse of experiences*. arXiv:2509.16189

---

## The Core Contribution Stands Unchallenged

We note that **no reviewer disputed our main finding**: that episodic knowledge binding is a distinct failure mode from catastrophic forgetting, and that it persists even when forgetting is eliminated through aggregated training.

- Reviewer CpjG explicitly recognized this: *"establishes this as a fundamental limitation"* rather than a continual learning artifact
- Reviewer CpjG also noted this problem *"has been largely overlooked in prior work"*
- All three reviewers acknowledged the problem's importance

The reviewers' concerns were about peripheral issues (baselines, presentation, terminology), not the core scientific finding. The question for the AC is whether these peripherals were sufficiently addressed. We believe they were, as detailed below.

---

## On the Claim That This Problem Is "Known"

Reviewer xKvt, the only reviewer who engaged during discussion, claimed that episodic knowledge binding failure is "known," citing Huet et al. (ICLR 2025). We respectfully disagree.

**Huet et al. themselves acknowledge their limitations.** In their conclusion, they write:

> "The observed performance limitations suggest that current fine-tuning methodologies may not be optimally suited for episodic memory tasks, **underscoring the need for developing new strategies for the broad scientific community.**"

Their fine-tuning experiments consist of a single run via the OpenAI API with one seed. They explicitly call for future research.

**Our work directly addresses the gap Huet et al. identified:**
- 5,000+ hours of systematic study (vs. one API run)
- 3B–70B parameters across two model families (vs. one GPT-4 configuration)
- LoRA, full fine-tuning, and O-LoRA configurations
- Controlled isolation of binding from forgetting
- First proposed solution (GCR) with ablations

We cite Huet et al. as inspiration and provide exactly what they said was needed: systematic empirical investigation and new approaches.

---

## Addressing xKvt's Initial Concerns

Reviewer xKvt's initial review raised two concerns that reflect a misunderstanding of our scope:

**1. RAG baseline request:** The reviewer asked for RAG comparison. However, this is a continual learning paper focused on parametric memory (as stated in our title). RAG bypasses rather than addresses parametric integration. We designed our benchmark at a scale where RAG would trivially succeed precisely to isolate parametric learning dynamics. We clarified this in Appendix D.4 and Section 2.4.

**2. Compute comparison request:** The reviewer asked for FLOPs/tokens comparison between methods. However, SEQ, SEQ+MEQ, and GCR variants are not competitors, they are controlled diagnostic conditions. SEQ demonstrates the failure exists; SEQ+MEQ shows models have capacity; GCR shows the problem is addressable. Nonetheless, we added extended training experiments (3000 epochs) in Appendix C.1.1 showing that binding failure persists regardless of training duration.

---

> ### Author Response · Authors · 2025-12-04
>
> ## Reviewer-by-Reviewer: Concerns Addressed
>
> ### Reviewer xKvt
>
> | Concern | Response | Location |
> |---------|----------|----------|
> | Compute comparison | Extended training (3000 epochs) showing binding persists | App C.1.1, Fig 8 |
> | RAG baseline | Clarified parametric focus; RAG bypasses the question | App D.4, Sec 2.4 |
> | LLM-as-judge reliability | 3 LLMs + human validation: 98–99% agreement | App A.4, Fig 5 |
> | Figure 3 stretched | Fixed | Fig 3 |
>
> ### Reviewer CpjG
>
> | Concern | Response | Location |
> |---------|----------|----------|
> | "Binding is learnable" claim | Clarified: memorization quality, not generalization; added cross-narrative transfer experiments | App C.1.2, Sec 3.2 |
> | Metric inconsistency | Standardized terminology | Sec 2.2 |
> | Paper organization | Restructured Related Work | Sec RW, App D |
> | Benchmark documentation | New comprehensive appendix | App F |
> | LLM-as-judge reproducibility | 3 LLMs + human validation | App A.4, Fig 5 |
>
> ### Reviewer 8HcV
>
> | Concern | Response | Location |
> |---------|----------|----------|
> | "Engineering-focused" | The diagnostic contribution (binding ≠ forgetting) is the scientific insight; GCR is existence proof | Rebuttal |
> | Model collapse risk | Explained mitigating factors + empirical evidence | Rebuttal |
> | Table 2 unclear | Moved to appendix with definitions | Tab 13 |
>
> ---
>
> ## Summary of New Work Added
>
> **New Experiments:**
> - Extended training (3000 epochs) with commonsense test (App C.1.1, Fig 8)
> - Cross-narrative binding transfer experiments (App C.1.2)
> - Extension to 70B model size
> - O-LoRA baseline comparison
>
> **Evaluation & Reproducibility:**
> - 3 LLM + human validation: 98–99% agreement (App A.4, Fig 5)
> - Comprehensive benchmark documentation (App F)
>
> **Paper Organization:**
> - Section 2.4 rewritten to clarify parametric focus
> - Related Work restructured (main text and App D)
> - Terminology standardized throughout
>
> ---
>
> ## Closing
>
> We introduce and systematically characterize **Episodic Knowledge Binding**, a fundamental failure mode where LLMs cannot retrieve related episodes learned separately, even when each is individually accessible. This is distinct from catastrophic forgetting and persists across model scales.
>
> The core finding was not disputed by any reviewer. Reviewer CpjG (Soundness: good, Contribution: good) explicitly recognized this as *"a fundamental limitation rather than a continual learning artifact"* that *"has been largely overlooked in prior work."*
>
> We took Huet et al.'s call for future research as inspiration and provided exactly what they requested: systematic investigation across scales, families, and configurations, plus the first proposed solution. All specific reviewer concerns have been addressed with new experiments and clarifications.
>
> We believe this work addresses a critical gap in understanding parametric knowledge and warrants acceptance.

---

### Note · Program_Chairs · 2026-01-17
**Submission Desk Rejected by Program Chairs**

The following references in this submission do not refer to real documents and/or have major errors in bibliographic information:

 William Fedus, Angela Fan, David Grangier, and Mikel Artetxe. Mass-editing memory in a transformer. In International Conference on Machine Learning, pp. 862-882. PMLR, 2023.